# Generation of human hepatic progenitor cells with regenerative and metabolic capacities from primary hepatocytes

**Takeshi Katsuda[1†], Juntaro Matsuzaki[1], Tomoko Yamaguchi[1,2], Yasuhiro Yamada[3], Marta Prieto-Vila[1], Kazunori Hosaka[1], Atsuko Takeuchi[4], Yoshimasa Saito[2], Takahiro Ochiya[1,5]***

[1]Division of Molecular and Cellular Medicine, National Cancer Center Research Institute, Tokyo, Japan; [2]Division of Pharmacotherapeutics, Faculty of Pharmacy, Keio University, Tokyo, Japan; [3]Department of Clinical Pharmaceutics, Nihon Pharmaceutical University, Saitama, Japan; [4]Division of Analytical Laboratory, Kobe Pharmaceutical University, Kobe, Japan; [5]Institute of Medical Science, Tokyo Medical University, Tokyo, Japan

**Abstract** Hepatocytes are regarded as the only effective cell source for cell transplantation to treat liver diseases; however, their availability is limited due to a donor shortage. Thus, a novel cell source must be developed. We recently reported that mature rodent hepatocytes can be reprogrammed into progenitor-like cells with a repopulative capacity using small molecule inhibitors. Here, we demonstrate that hepatic progenitor cells can be obtained from human infant hepatocytes using the same strategy. These cells, named human chemically induced liver progenitors (hCLiPs), had a significant repopulative capacity in injured mouse livers following transplantation. hCLiPs redifferentiated into mature hepatocytes in vitro upon treatment with hepatic maturation-inducing factors. These redifferentiated cells exhibited cytochrome P450 (CYP) enzymatic activities in response to CYP-inducing molecules and these activities were comparable with those in primary human hepatocytes. These findings will facilitate liver cell transplantation therapy and drug discovery studies.

**\*For correspondence:** tochiya@tokyo-med.ac.jp

**Present address:** [†]Perelman School of Medicine, University of Pennsylvania, Philadelphia, United States

## Introduction

Expansion of functional human hepatocytes is a prerequisite for liver regenerative medicine. Human hepatocytes are currently regarded as the only competent cell source for transplantation therapy (*Fisher and Strom, 2006*); however, their availability is limited due to a shortage of donors. Moreover, the therapeutic application of hepatocytes is hampered by their inability to proliferate in vitro. To overcome this, researchers have sought to generate expandable cell sources as alternatives to primary hepatocytes. Such cell sources include embryonic stem cell- and induced pluripotent stem-cell-derived hepatic cells (*Carpentier et al., 2014*; *Liu et al., 2011*; *Takebe et al., 2013*; *Woo et al., 2012*; *Zhu et al., 2014*), lineage-converted hepatic cells (induced hepatic cells; (*Du et al., 2014*; *Huang et al., 2014*), and facultative liver stem/progenitor cells (LPCs) residing in adult liver tissue (*Huch et al., 2015*). However, while primary hepatocytes efficiently repopulate injured mouse livers (repopulation indexes (RIs) > 50%), the repopulation efficiency of these laboratory-generated hepatocytes is limited, with reported RIs generally less than 5% (reviewed in *Rezvani et al., 2016*).

Researchers have also attempted to expand primary human hepatocytes (PHHs) in vitro. Several studies reported the expansion of these cells (*Hino et al., 1999*; *Shan et al., 2013*; *Utoh et al., 2008*; *Walldorf et al., 2004*; *Yamasaki et al., 2006*), suggesting that they are potentially applicable for transplantation therapy. However, the growth rate and proliferative lifespan of PHHs are limited.

**eLife digest** One of the most successful treatments for liver disease is transplanting a donor liver into a patient. But demands for donor livers far outstrips supply. A promising alternative could be, rather than replacing the whole organ, to transplant patients with individual liver cells called hepatocytes. These cells can then move into the liver, replace damaged cells, and help support the organ. However, hepatocytes are also in short supply, as despite the liver's amazing regenerative abilities, these cells struggle to divide outside of the body. Improving how these cells multiply, could therefore help more people receive hepatocyte transplants.

In 2017, researchers found a way to convert mouse and rat hepatocytes into cells that could divide more rapidly using a cocktail of three small molecules. These 'chemically induced liver progenitors', or CLiPs for short, were able to mature into working hepatocytes and support injured mouse livers. But, discoveries made in rats and mice are not always applicable to humans. Now, Katsuda et al. – including some of the researchers involved in the 2017 work – have set out to investigate whether CLiPs can also be made from human cells, and if so, whether these cells can be used for hepatocyte transplantations.

Using a similar cocktail of molecules, Katsuda et al. managed to convert infant human hepatocytes into CLiPs. As with the rodent cells, these human CLiPs were able to turn back into mature, working liver cells. When transplanted into mice with genetic liver diseases, the human CLiPs moved into the liver and became part of the organ. These transplanted cells were able to reconstruct the liver tissue of diseased mice, and in some cases, replaced more than 90% of the liver's damaged cells.

Developing human CLiP technology could provide a new way to support people on the waiting list for liver transplantation. But there are some obstacles still to overcome. At present the technique only works with hepatocytes from infant donors. The next step is to improve the method so that it works with liver cells donated by adults.

For example, Yoshizato's group reported that PHHs can be cultured for several passages, but their growth rate is slow (population doubling time of 20–300 days) (*Yamasaki et al., 2006*). This finding indicates that culture of PHHs must be improved for the clinical application of these cells.

We recently reported that a cocktail of small molecule signaling inhibitors reprograms rodent adult hepatocytes into culturable LPCs, named chemically induced liver progenitors (CLiPs) (*Katsuda et al., 2017*). Notably, rat CLiPs extensively repopulate chronically injured mouse livers without causing any tumorigenic features. Here, using the same strategy, we demonstrate that human infant hepatocytes can be also converted into proliferative LPC-like cells, which are named human CLiPs.

## Results

### Small molecules support expansion of PHHS

In a pilot study, we tested whether the combination of Y-27632 (Y), A-83–01 (A), and CHIR99021 (C), the chemical cocktail used to reprogram rodent hepatocytes, also induced proliferation of commercially available cryopreserved adult PHHs (APHHs) (donor information is summarized in (*Table 1*). In contrast with the basal culture medium (small hepatocyte medium (SHM)), culture in YAC-containing SHM (SHM+YAC) induced the proliferation of cells that morphologically resembled epithelial cells (*Figure 1A*). These cells were small and had a higher nucleus-to-cytoplasm ratio than hepatocytes, which is a typical morphological feature of LPCs. When colonies became densely packed, rat and mouse CLiPs exhibited a compact polygonal cell shape delimited by sharply defined refractile borders with bright nuclei in phase contrast images (*Figure 1B and C*). However, unlike rat and mouse CLiPs, the morphology of human cells did not clearly change after colonies became densely packed (*Figure 1A*). Although we did not perform further characterization, these proliferating cells likely arose from non-hepatic cells, such as biliary epithelial cells (BECs) or so-called liver epithelial cells, the origins of which are not well-defined (*Mitaka et al., 1999*). Thus, we speculated that human hepatocytes require additional proliferative stimuli. Therefore, we tested the ability of fetal bovine serum

**Table 1.**
Donor information of primary human hepatocytes (PHHs) used in this study.

| Cell type | IPHH | IPHH | IPHH | IPHH | IPHH | APHH | APHH | APHH | APHH | APHH |
|---|---|---|---|---|---|---|---|---|---|---|
| Lot | FCL | DUX | JFC | MRW | 187273 | HC7-4 | HC5-25 | HC1-14 | HC3-14 | 187271 |
| Age | 10 mo | 8 mo | 1 yr | 11 mo | 2 yr | 7 yr | 56 yr | 55 yr | 45 yr | 26 yr |
| Sex | Female | Male | Male | Male | Male | Male | Male | Male | Male | Male |
| Race | Hispanic | Caucasian | Caucasian | Caucasian | Caucasian | Caucasian | Caucasian | Caucasian | Caucasian | Caucasian |
| Cause of death | Anoxia/ drowning | Anoxia/ cardiovascular | Anoxia/ second to blunt injury | Asphyxiation | N/A | Anoxia | Cerebrovascular Accident | Anoxia | Cerebrovascular Accident | N/A |
| CMV | - | - | - | + | N/A | + | + | - | - | N/A |
| HIV | - | - | - | - | - | - | - | - | - | - |
| HBV | - | + | + | - | - | - | - | - | - | - |
| HCV | - | - | - | - | - | - | - | - | - | - |
| EBV | - | N/A | N/A | N/A | N/A | N/A | N/A | N/A | N/A | N/A |
| RPR | - | N/A | - | - | N/A | N/A | N/A | N/A | N/A | N/A |
| HTLV | N/A | N/A | N/A | - | N/A | N/A | N/A | - | - | N/A |

IPH: Infant primary human hepatocyte; APH: adult primary human hepatocyte; CMV: cytomegarovirus; HIV: human immunodeficiency virus; HBV: hepatitis B virus; HCV: hepatitis C virus; EBV: Epstein-Barr virus; RPR: rapid plasma reagin; HTLV: human T-cell leukemia virus; N/A: information not available.

(FBS) to support the proliferation of these cells. One of three lots of APHHs formed proliferative and

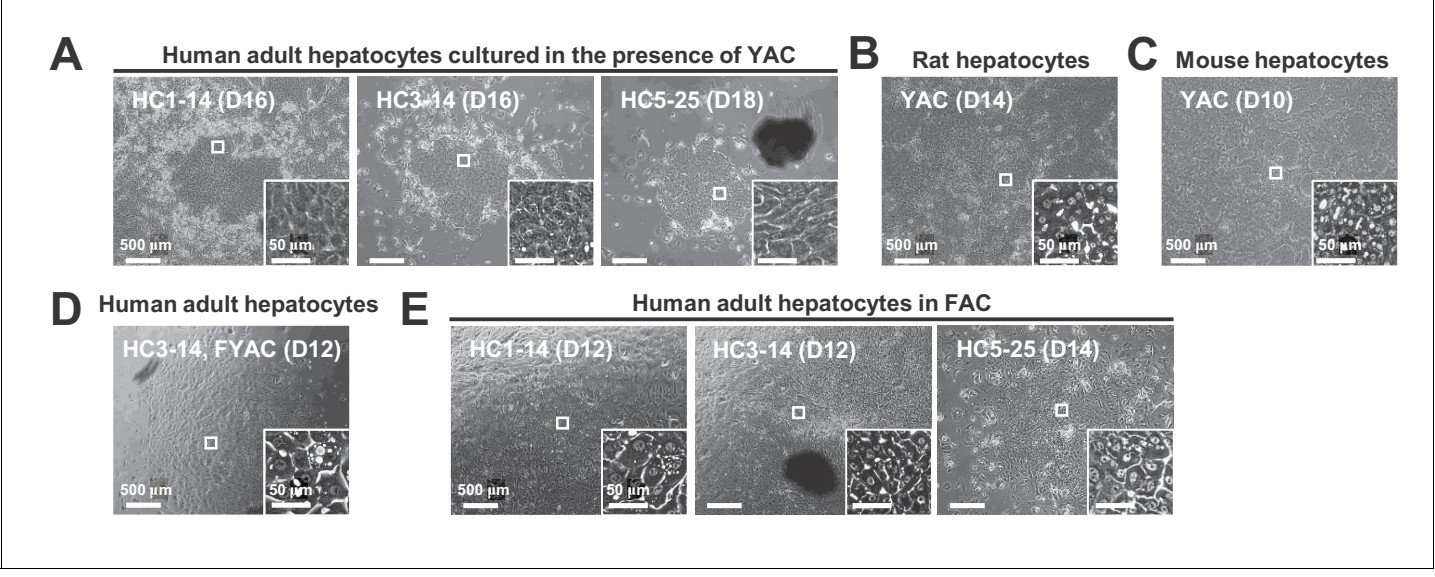

**Figure 1.** Morphological changes of hepatocytes in response to small molecule stimuli with/without FBS. (**A**) Phase contrast images of APHHs cultured in the presence of YAC, which are used to obtain rat and mouse CLiPs. Insets indicate representative magnified images. (**B**) Phase contrast images of rat CLiPs obtained by culture in the presence of YAC. The inset shows cells that spontaneously differentiated into mature hepatocyte (MH)-like cells in densely packed regions. (**C**) Phase contrast images of mouse CLiPs obtained by culture in the presence of YAC. The inset shows cells that spontaneously differentiated into MH-like cells in densely packed regions. (**D**) Phase contrast images of APHHs cultured in the presence of YAC and 10% FBS (FYAC). The inset shows cells that spontaneously differentiated into MH-like cells in densely packed regions. (**E**) Phase contrast images of APHHs cultured in FAC. Insets show cells that spontaneously differentiated into MH-like cells in densely packed regions.

densely packed colonies, and exhibited a hepatocytic morphology upon culture in medium supplemented with YAC and 10% FBS (FYAC) (*Figure 1D*). By contrast, all three lots of APHHs formed proliferative colonies with hepatic morphologies upon culture in medium supplemented with AC and 10% FBS (FAC) (*Figure 1E*). However, the proliferative capacity of these hepatic colony-forming cells was limited, and the number of these cells markedly decreased after the first passage, while non-parenchymal cells (NPCs) with non-hepatic morphologies became the dominant population (data not shown).

Next, considering the previous finding that PHHs derived from young donors are optimal for in vitro expansion (*Walldorf et al., 2004*; *Yamasaki et al., 2006*), we tested whether infant PHHs (IPHHs) expanded more efficiently in the presence of small molecules and FBS. Using IPHHs derived from a 10-month-old donor (lot FCL), we performed a mini-screen using all possible combinations of Y, A and C in 10% FBS-supplemented SHM. The water-soluble tetrazolium salt-based (WST) assay demonstrated that these cells proliferated in the presence of A, YA, AC and YAC (*Figure 2A*). Consistent with the observations made in APHHs (*Figure 1E*), these cells proliferated most efficiently in FAC and thus we used this medium in all subsequent experiments. Robust proliferation of hepatocytes was not supported by culture in the presence of AC or FBS alone, but was synergistically supported by culture in the presence of both AC and FBS (*Figure 2B*). Although proliferating cells cultured in FAC did not morphologically resemble hepatocytes when the cell density was low, they spontaneously acquired a hepatocyte-like morphology as colonies became densely packed (*Figure 2C*). This observation strongly suggests that human proliferative cells cultured in FAC more closely resembled rodent CLiPs than those cultured in the presence of YAC. Unlike APHHs, IPHHs proliferated efficiently and became the predominant population over 2 weeks of culture. Two other lots of IPHHs (lot DUX from an 8-month-old donor and lot JFC from a 1-year-old donor) (Table 1) also proliferated in this culture condition, although the proliferative capacity varied among the lots: FCL, DUX and JFC proliferated $49.2 \pm 9.34$ (at day 14), $46.2 \pm 2.12$ (at day 14) and $3.66 \pm 0.321$ (at day 12) folds, respectively (mean $\pm$ SEM, determined by two repeated experiments for each lot). We also confirmed by microscopy that FAC enabled two more donors (11 months and 2-year-old)-derived IPHHs and one juvenile donor (7-year-old)-derived hepatocytes to proliferate and spontaneously change their morphologies to hepatocyte-like ones in the densely packed region of the proliferating colonies (*Figure 2—figure supplement 1*).

## Characterization of proliferating cells cultured in FAC

These proliferating cells expressed multiple surface markers of LPCs, including EPCAM, CD44, PROM1 (also known as CD133), CD24, and ITGA6 (*Figure 2D*, *Figure 2—figure supplement 2A*). It should be noted that PHHs before plating minimally expressed these LPC markers (*Figure 2—figure supplement 2B*). Although we used PHHs which underwent a freeze-thaw cycle, we confirmed that expression of DPP4/CD26, a general hepatocyte marker, was still preserved, supporting the validity of this flow cytometry analysis. We observed that PHHs were slightly positive for ITGA6, but this is likely to be a nature of primary hepatocytes, because lineage-traced mouse primary hepatocytes also exhibited a slight signal of Itga6, while none of the other analyzed LPC markers, Epcam, Prom1 and Cd24 were detected (*Figure 2—figure supplement 2C*).

Next, we performed microarray-based transcriptome analysis of previously identified BEC/LPC marker genes to further characterize these cells. Expression of many of these genes was induced during the 2 weeks of culture (*Figure 2E*). Some of these genes, such as *PROM1* and *SPP1*, were expressed at comparable levels regardless of whether cells were cultured in the presence of AC, suggesting that their expression was spontaneously induced by the basal culture conditions (*Figure 2E*). However, expression of multiple BEC/LPC marker genes, including *EPCAM*, *SOX9*, *KRT19*, *TACSTD2*, *AXIN2* and *PROX1*, was increased in cells cultured in FAC (*Figure 2E and F*). Of these, expression of *EPCAM*, *SOX9*, and *KRT19* was affected not only by the presence of AC but also by the culture duration, suggesting that AC-induced expression of these genes during in vitro culture. By contrast, expression of *AXIN2* and *PROX1* was maintained, but not increased, upon culture in the presence of AC. Gene signature enrichment analysis (GSEA) comparing cells cultured in the presence of FBS and those cultured in FAC demonstrated that the majority of gene sets enriched in the latter cells were related to hepatic function (*Figure 2G*, *Supplementary file 1*), suggesting that AC also helped to maintain the hepatocytic characteristics of cultured hepatocytes. Although cell-cycle-related gene sets were also identified by GSEA, their enrichment scores were relatively

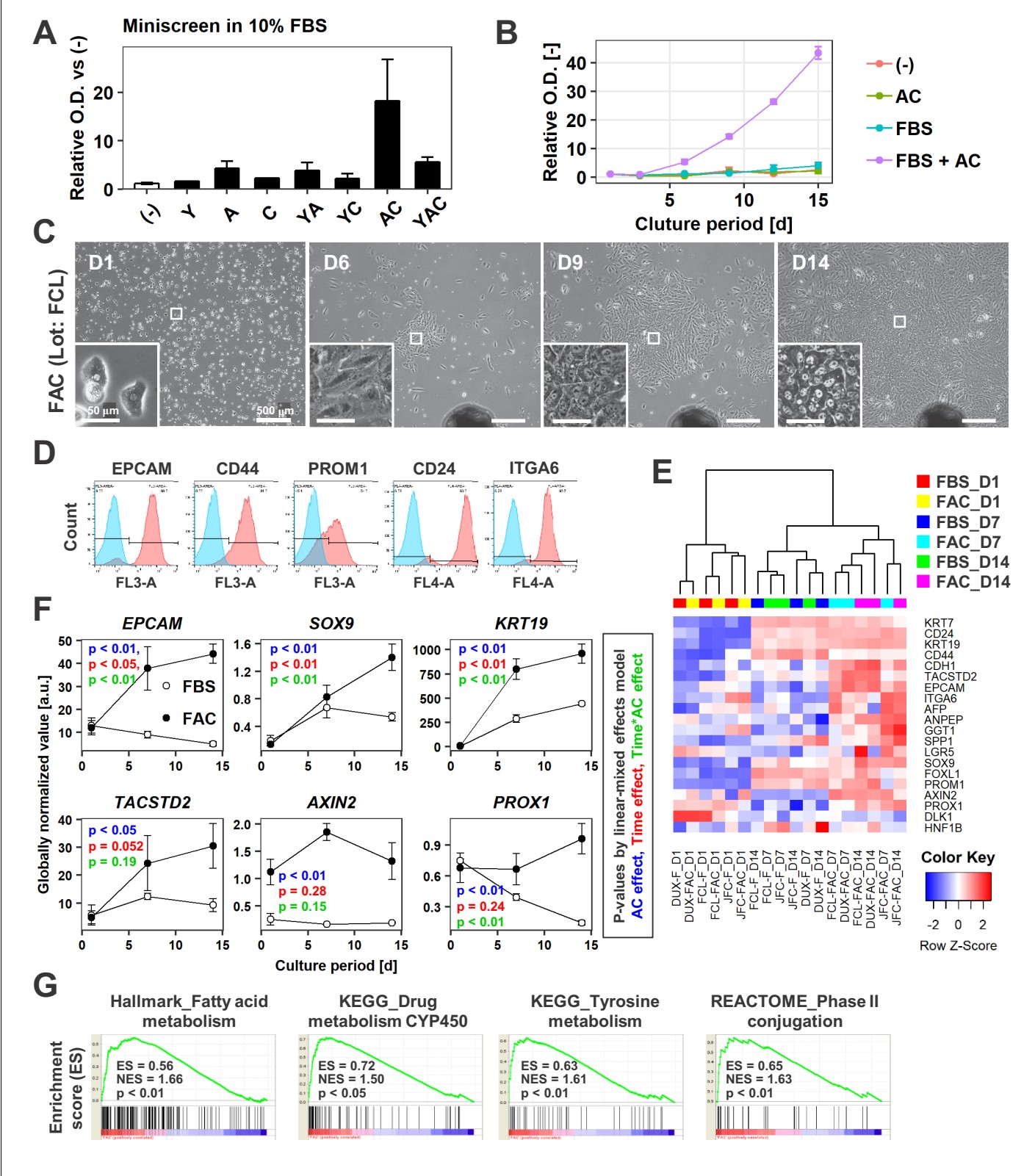

**Figure 2.** AC together with FBS support the expansion of IPHHs. (**A**) WST assay assessing the effects of various combinations of Y, A, and C together with 10% FBS on proliferation of 8-month-old IPHHs (lot FCL). Absorbance at 450 nm was determined at D14 and normalized against that at D0. Data are the mean ± SEM of two repeated experiments. (**B**) WST assay assessing the effects of AC and FBS on proliferation of IPHHs (lot FCL). Absorbance at 450 nm was determined at D14 and normalized against that at D0. Data are the mean ± SD of three technical replicates. (**C**) Phase contrast images

*Figure 2 continued on next page*

*Figure 2 continued*

showing the morphological changes of IPHHs (lot FCL) upon culture in FAC. Inset images show spontaneous hepatic differentiation in densely packed regions at D14. (D) Flow cytometric analysis of surface expression of LPC markers. Results of cells from lot FCL are shown as representative data (see also *Figure 2—figure supplement 2A*). (E) Heatmap showing expression of BEC/LPC marker genes, as assessed by microarray analysis. Each element represents normalized (log2) expression, as indicated by the color scale. Data are from three lots and two repeated experiments. Hierarchical clustering was performed based on Euclidean distance. (F) Expression levels of genes that were differentially expressed between cells cultured in the presence of FBS and those cultured in FAC are shown as mean ± SEM of three lots per time point (each value is determined as the mean of 2 repeated experiments for each lot). p-Values were calculated by the linear mixed model to account for the covariance structure due to repeated measures at different time points. The meanings of the various colors are described in the figure. (G) GSEA demonstrating enrichment of hepatic function-related gene sets in cells cultured in FAC in comparison with cells cultured in the presence of FBS at D14. p-Values indicate nominal p-values.

The online version of this article includes the following figure supplement(s) for figure 2:

**Figure supplement 1.** Proliferative colonies observed in infant and juvenile PHHs.
**Figure supplement 2.** Characterization of FAC-cultured proliferative human hepatic cells.
**Figure supplement 3.** Cell-cycle-related gene expression analysis by GSEA.

---

low (*Figure 2—figure supplement 3A*, *Supplementary file 1*). This is likely because cell proliferation was also increased in part by culture in FBS alone. Indeed, proliferation-related gene sets were enriched both in cells cultured in FBS only and in FAC compared with D1 hepatocytes (*Figure 2— figure supplement 3B and C*, *Supplementary file 2*, *3*). In summary, two small molecules, AC, together with FBS, support the proliferation of hepatic epithelial cells with characteristics of both hepatocytes and LPCs/BECs.

## Comparison of IPHHs and APHHs in terms of their responsiveness to FAC

To investigate the difference regarding the responsiveness to FAC of IPHHs and APHHs, we compared their transpcriptome by microarray analysis. Hierarchical clustering of the whole transcriptome demonstrated that IPHHs cultured in FAC for 7 or 14 days formed a cluster distinct from those cultured in FBS (*Figure 3A*). In contrast, APHHs cultured in FAC for 7 or 14 days were not clearly separated from those cultured in FBS. These results suggest that APHHs are less sensitive to AC than IPHHs. GSEA indicated that many of the signaling pathways enriched for IPHHs cultured in FAC for 7 days compared with APHHs were cell-cycle-related pathways (*Figure 3—figure supplements 1* and *2*) (we avoided comparing cells at D14, because lot 187271 APHHs were severely contaminated with NPCs at D14, as shown in *Figure 3—figure supplement 3*, (iv)). In contrast, pathways enriched for APHHs included hepatic function-associated ones (*Figure 3—figure supplements 1* and *2*). These results suggest that APHHs were not susceptible to the pro-proliferative effect of AC. Intriguingly, we observed a relatively similar expression profile of LPC marker genes between IPHHs and APHHs (*Figure 3B*), except for *EPCAM* and *ANPEP*. We then asked whether APHHs indeed responded to A83-01 and CHIR99021. GSEA indicated enrichment of Wnt signaling in IPHHs compared with APHHs (nominal p-value=0.019), suggesting that APHHs less efficiently responded to CHIR99021 (*Figure 3—figure supplements 1* and *4*). In contrast, to our surprise, TGFβ signaling was enriched in IPHHs compared with APHHs (nominal p-value<0.001) (*Figure 3—figure supplements 1* and *4*). We further investigated the expression of individual genes which are known to be in the downstream of Wnt signaling (*Russell and Monga, 2018*) and TGFβ signaling (*Cicchini et al., 2015*; *Fabregat and Caballero-Díaz, 2018*) (*Figure 3C and D*). Both IPHHs and APHHs upregulated typical Wnt target genes in hepatocytes, such as *GLUL* and *CYP1A2* in the presence of AC (*Figure 3C*). On the other hand, we found that LPC-related Wnt-target genes, *AXIN2* and *LGR5*, were expressed at higher levels in IPHHs than APHHs (*Figure 3C*). We also confirmed that TGFβ downstream genes were downregulated in both IPHHs and APHHs treated with FAC compared with their FBS counterparts (*Figure 3D*). In addition, we confirmed that expression levels of some of these genes, for example *VIM*, *SNAI1* and *ZEB1*, were higher in IPHHs than APHH, which explains the reason for the enrichment of TGFβ signaling in IPHHs by GSEA (*Figure 3—figure supplements 1* and *4*). Another signaling pathway enriched in IPHHs was mTORC1 signaling (*Figure 3—figure supplement 4*). mTORC1 is activated specifically in pericentral hepatocytes in a Wnt signaling-dependent manner, and suggested to regulate their growth in normal liver (*Adebayo Michael et al., 2019*). mTORC1 is also reported to be essential for BEC expansion during ductular reaction in regenerating

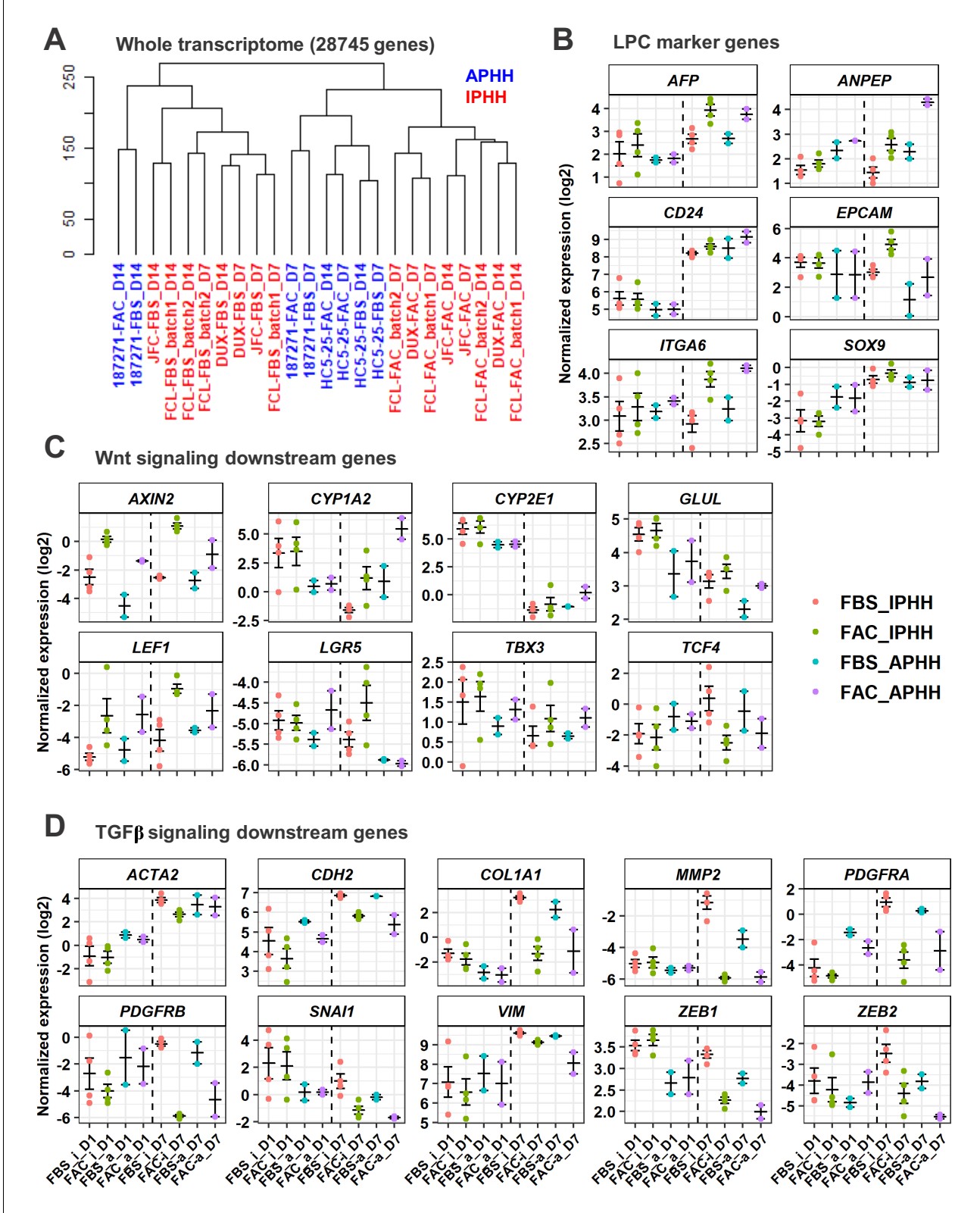

**Figure 3.** Comparative analysis of responsiveness to FAC of IPHHs and APHHs. (**A**) Hierarchical clustering for whole transcriptome of IPHHs (red) and APHHs (blue). Two lots of APHHs (HC5-25 and 187271) and three lots of IPHHs were compared. (**B**) Gene expression of LPC marker genes was compared between IPHHs and APHHs at D1 and D7 of culture in FBS or FAC. Data are the mean ± SEM. (**C**) Gene expression of Wnt signaling downsteam genes was compared between IPHHs and APHHs at D1 and D7 of culture in FBS or FAC. Data are the mean ± SEM. (**D**) Gene expression of

*Figure 3 continued on next page*

Figure 3 continued

TGFβ signaling downsteam genes was compared between IPHHs and APHHs at D1 and D7 of culture in FBS or FAC. Data are the mean ± SEM. FBS_i, FAC_i, FBS_a and FAC_a below each panel denote IPHHs cultured in FBS and FAC, and APHHs cultured in FBS and FAC, respectively.

The online version of this article includes the following figure supplement(s) for figure 3:

**Figure supplement 1.** GSEA comparing IPHHs and APHHs cultured in FAC for 7 days (KEGG database).

**Figure supplement 2.** GSEA comparing IPHHs and APHHs cultured in FAC for 7 days (Hallmark database).

**Figure supplement 3.** Morphological changes of APHHs and IPHHs which were cultured in FAC medium.

**Figure supplement 4.** Individual plots of GSEA results for TGFβ, Wnt and mTORC1 signaling pathways.

liver as well as BEC organoid formation in vitro (*Planas-Paz et al., 2019*). In summary, although a more detailed analysis is needed, the low proliferative capacity of APHHs might be partly explained by their lower responsiveness to Wnt signaling.

## Hepatic differentiation capacity of the proliferative cells

A hepatic differentiation capacity is an important feature of LPCs, particularly for their potential use as a candidate cell source for transplantation therapy. To investigate the hepatic differentiation capacity of these proliferative cells, we passaged and cultured them in the presence of oncostatin M (OSM), dexamethasone and Matrigel, which induce maturation of LPCs into hepatocytes (*Kamiya et al., 2002*). As noted in *Figure 2C*, the proliferative cells spontaneously acquired hepatic morphologies when they reached 100% confluency, even in the absence of hepatic maturation inducers (*Figure 4A*, *Figure 4—figure supplement 1A*, middle panels for each lot). However, this morphological change was more evident in the presence of hepatic maturation inducers (*Figure 4A*, *Figure 4—figure supplement 1A*, right panels for each lot). In particular, cells acquired a polygonal and cytoplasm-rich morphology, which is similar to that of PHHs (*Figure 4B*). Accordingly, microarray analysis confirmed that expression of representative hepatic marker genes, including *ALB*, *TDO2* and *SERPINA1* was increased after hepatic maturation induction (*Figure 4C*). However, the expression levels of these genes were not markedly changed in cells from lot JFC. This is presumably because expression of hepatic maturation genes was already high in these cells even before hepatic induction. In contrast with the hepatic marker genes, expression of the BEC/LPC marker genes including *SOX9*, *KRT19*, and *KRT7* was decreased, suggesting that the proliferative cells lost their BEC/LPC phenotype and acquired a mature hepatic phenotype (*Figure 4—figure supplement 1B*). Hierarchical cluster analysis of genes that were differentially expressed between cells cultured in the presence of hepatic maturation inducers (Hep-i(+)) and cells cultured for the same duration in the absence of hepatic maturation inducers (Hep-i(-)) indicated that the characteristics of Hep-i(+) cells were relatively similar to those of PHHs (*Figure 4D*). Overrepresented pathways in Hep-i(+) cells in comparison with Hep-i(-) cells were associated with the immune response and metabolic processes (*Figure 4E*), both of which are important functions of the liver. These findings were further validated by GSEA (*Figure 4F*, *Supplementary file 4*). By contrast, overrepresented pathways in Hep-i(-) cells in comparison with Hep-i(+) cells were associated with developmental processes and morphogenesis, implying that Hep-i(-) cells were functionally immature compared with Hep-i(+) cells (*Figure 4—figure supplement 1C*). In addition, cell cycle-related genes were overrepresented in Hep-i(-) cells (*Figure 4—figure supplement 1D*, *Supplementary file 5*), which is consistent with the general notion that progenitor cells have a greater proliferative capacity than cells with a more mature phenotype. Taken together, proliferative cells derived from human hepatocytes via culture in FAC lost their immature phenotype and acquired a mature hepatocyte-like phenotype in response to hepatic maturation inducers. Thus, we hereafter designate these proliferative cells as human CLiPs (hCLiPs).

## Expression and activities of drug-metabolizing enzymes in hCLiP-derived hepatocytes

Cytochrome P-450 (CYP) enzymes play a central role in the metabolic functions of the liver. Thus, we investigated the metabolic functions of hCLiP-derived hepatocytes. As noted in the previous section, overrepresented pathways in Hep-i(+) cells were associated with metabolism (*Figure 4E and F*, *Supplementary file 5*). In addition, pathways involving CYPs were enriched in Hep-i(+) cells, as characterized by GSEA using both the KEGG and Reactome databases, although the p-values for these

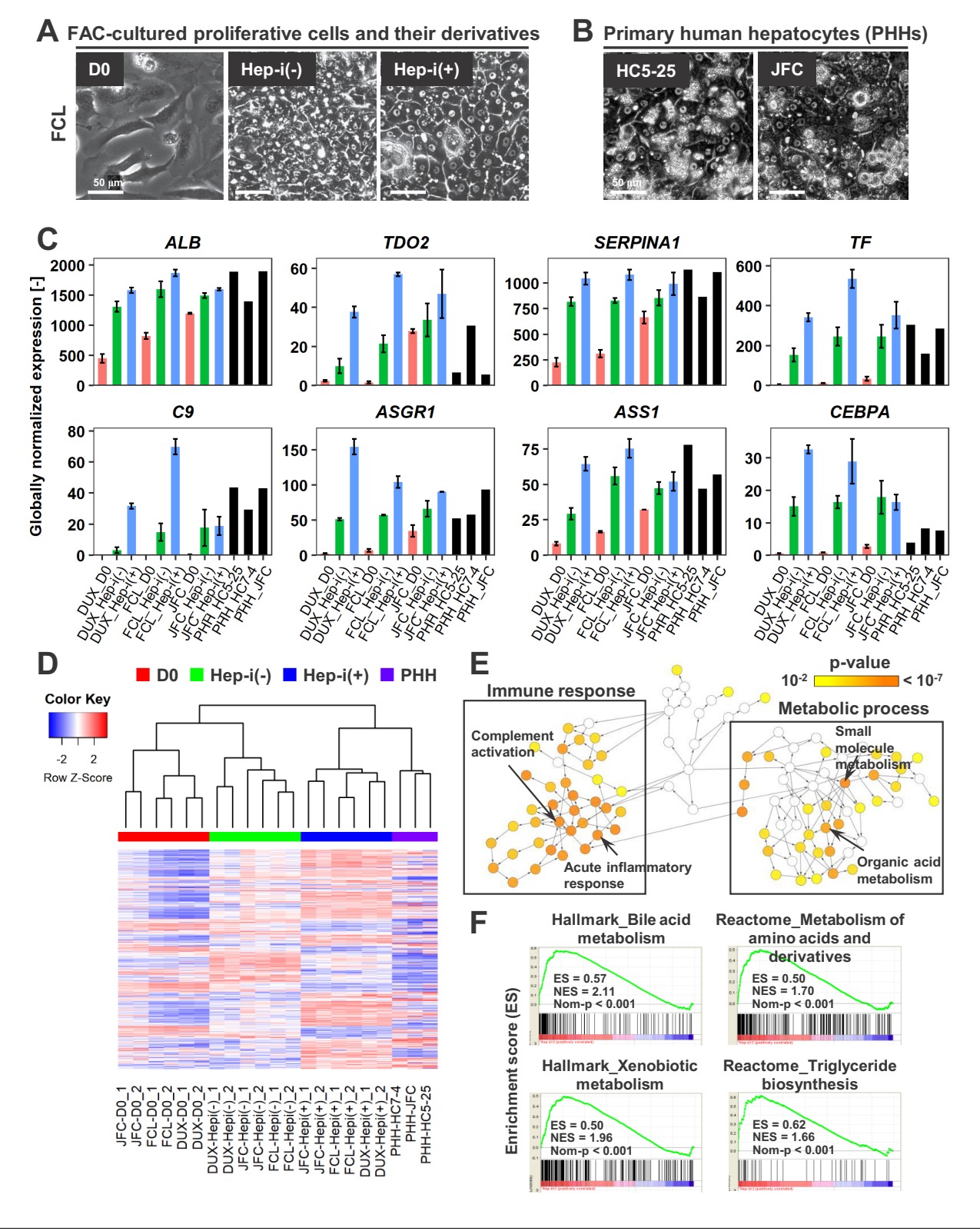

**Figure 4.** FAC-cultured proliferative cells differentiate into mature hepatocytes in vitro. (**A**) Phase contrast images showing the morphological changes of FAC-cultured human proliferative cells (lot FCL) treated with (Hep-i(+)) or without (Hep-i(-)) hepatic maturation-inducing factors. Also see *Figure 4—figure supplement 1A* for lots DUX and JFC. (**B**) Phase contrast images of PHHs for reference. (**C**) Quantified expression of hepatic function-related genes in hCLiPs derived from the three lots with or without hepatic induction and in PHHs. Data are shown as mean ± SEM of two repeated

*Figure 4 continued on next page*

*Figure 4 continued*

experiments for each lot of hCLiPs and the results of one experiment for each lot of PHHs. (D) Hierarchical clustering based on Canberra distance of 990 genes that were differentially expressed (≥2 fold change on average for the three lots and p<0.05 by the paired t-test) between Hep-i(-) and Hep-i(+). Data were obtained from two repeated experiments for each lot of hCLiPs and from one experiment for each lot of PHHs. (E) Biological processes overrepresented in Hep-i(+) cells in comparison with Hep-i(-) cells, as identified using BiNGO, a Cytoscape plug-in. p-value is calculated by the default setting of the plug-in. (F) GSEA demonstrating enrichment of hepatic function-related gene sets in Hep-i(+) cells in comparison with Hep-i(-) cells. p-Values indicate nominal p-values.

The online version of this article includes the following figure supplement(s) for figure 4:

**Figure supplement 1.** Characterization of proliferative human hepatic cells following hepatic maturation.

gene sets were not lower than 0.05 (*Figure 4—figure supplement 1E*). A heatmap revealed that expression of several *CYP* genes was higher in Hep-i(+) cells than in Hep-i(-) cells (*Figure 5A*). These genes included *CYP2B6*, *CYP2D6*, *CYP2E1*, *CYP2C9* and *CYP3A4*, which play crucial roles in metabolic functionality of the human liver (*Martignoni et al., 2006*). The enzymatic activities of multiple CYPs were investigated by liquid chromatography tandem mass spectrometry (LC-MS/MS) using a cocktail of substrates (*Figure 5B*) (*Ohtsuki et al., 2012*). This revealed that the enzymatic activities of CYP1A2, CYP2C19, CYP2C9, CYP2D6 and CYP3A were comparable, if not the same, in Hep-i(+) cells derived from lots FCL and JFC as in PHHs, but were lower in Hep-i(+) cells derived from lot DUX (*Figure 5B*). Expression of CYP1A2, CYP2B6 and CYP3A4 is induced in hepatocytes via transcriptional activation in response to certain chemicals. Thus, we investigated whether the expression and activities of these CYPs were increased in hCLiP-derived hepatocytes treated with prototypical inducers of each CYP isoform, namely, omeprazole (aryl hydrocarbon receptor ligand) for CYP1A2, phenobarbital (indirect activator of constitutive active androstane receptor) for CYP2B6 and CYP3A4, and rifampicin (pregnane X receptor ligand) for CYP3A4. These *CYP* genes were markedly upregulated in cells derived from the three lots in response to the corresponding inducer (*Figure 5—figure supplement 1A,B*). Although enzymatic activities of these CYPs were increased in both Hep-i(-) and Hep-i(+) cells upon treatment with the corresponding inducer, these increases were relatively larger in the latter cells (*Figure 5C*, *Figure 5—figure supplement 1C*), consistent with the changes in gene expression (*Figure 5—figure supplement 1B*). We also directly quantified CYP protein expression by mass spectrometry. Protein expression of CYP1A2 and CYP3A4 in hCLiP-derived hepatocytes was increased in response to the corresponding inducer (*Figure 5D*). In addition, activities of the phase II enzymes sulfotransferase (SULT) and UDP-glucuronosyltransferase (UGT) were comparable in hCLiP-derived hepatocytes and PHHs (*Figure 5E*). These results demonstrate that hCLiPs differentiate into cells that are metabolically mature after induction of hepatic maturation and thus are potentially applicable for drug metabolism studies.

## Long-term expansion of hCLiPs

Long-term culture of hepatocytes or LPCs with a sustained proliferative capacity is of great interest for liver regenerative medicine and drug discovery studies. Thus, we investigated the feasibility of long-term culture of hCLiPs. Cells derived from lots FCL and DUX could be serially passaged until at least passage 10 (P10) without growth arrest (*Figure 6A*) or obvious morphological changes (*Figure 6—figure supplement 1A*). The population doubling times of FCL and DUX hCLiPs were 1.27 ± 0.0066 and 1.43 ± 0.0086 d, respectively (mean ± SEM, determined by three repeated experiments for each lot). However, non-hepatic cells with a fibroblast-like morphology were also observed (*Figure 6—figure supplement 1A*, arrows), and the percentage of these cells varied among repeated experiments for each lot, as assessed by flow cytometric analysis of the epithelial-cell surface marker proteins EPCAM and CD24 (*Figure 6—figure supplement 1B*). Cultures of cells from lot JFC contained more fibroblast-like cells than cultures of cells from lots FCL and DUX (*Figure 6—figure supplement 1A*). Upon culture of cells from lot JFC, the percentage of fibroblastic cells increased with the passage number and fibroblastic cells overwhelmed hCLiPs by P5, as assessed by microscopic observation (n = 3 repeated experiments) (*Figure 6—figure supplement 1A*) and flow cytometric analysis of LPC markers (n = 1 experiment) (*Figure 6—figure supplement 1B*). However, when EPCAM⁺ cells were sorted from primary hCLiPs at the first passage, proliferative epithelial cells were observed for at least the next three passages (total of four passages) with their population

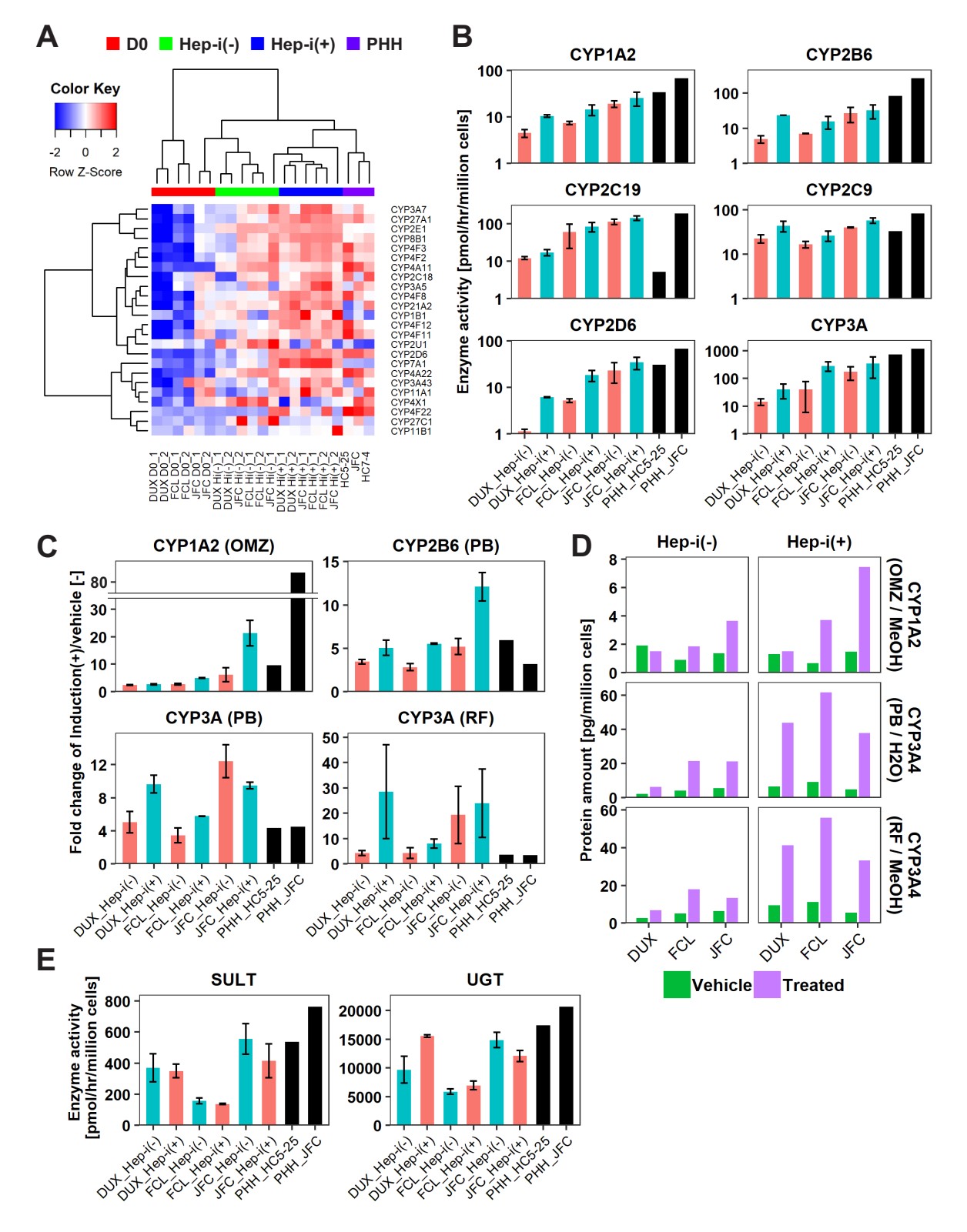

**Figure 5.** hCLiP-derived hepatocytes exhibit CYP enzymatic activity. (**A**) Heatmap showing expression of *CYP* genes that were differentially expressed between Hep-i(-) and Hep-i(+) cells (≥1.5 fold change), as assessed by microarray analysis. Fold change was calculated using the mean values of three donor-derived CLiPs (experiments were repeated twice for each donor-derived CLiPs). Hierarchical clustering was performed based on Euclidean distance. (**B**) Basal enzymatic activities of major CYPs in Hep-i(-) cells, Hep-i(+) cells, and PHHs, as assessed by LC-MS/MS using a cocktail of substrates. *Figure 5 continued on next page*

*Figure 5 continued*

Data were obtained from two repeated experiments for each lot of hCLiPs and from one experiment for each lot of PHHs. (C) Inducibility of CYP1A2, CYP2B6, and CYP3A activities. Enzymatic activities in inducer-treated cells were compared with those in cells treated with the corresponding vehicle by LC-MS/MS analysis using a cocktail of substrates. Data are the mean ± SEM of two repeated experiments for each lot of hCLiPs and the results of one experiment for each lot of PHHs. (D) LC-MS/MS analysis of the intracellular protein levels of CYP1A2 and CYP3A4 in Hep-i(-) and Hep-i(+) cells treated with inducers or the corresponding vehicle. Data are from one experiment for each lot of hCLiPs. (E) Enzymatic activities of the phase II enzymes UGT and SULT, as assessed by LC-MS/MS analysis using a cocktail of substrates. Data are the mean ± SEM of two repeated experiments for each lot of hCLiPs and the results of one experiment for each lot of PHHs.

The online version of this article includes the following figure supplement(s) for figure 5:

**Figure supplement 1.** Inducibility of CYP1A2 and CYP3A4 in Hep-i(+) cells.

doubling time 1.24 d (n = 1 experiment) between P1 and P4 (*Figure 6A*, *Figure 6—figure supplement 1A*), confirming the proliferative capacity of hCLiPs obtained from lot JFC. Although expression of surface markers varied among experimental batches at later passages (*Figure 6—figure supplement 1B*), it was relatively stable up to P5 in cells derived from lots FCL and DUX (*Figure 6—figure supplement 1B*). We also investigated the karyotype of cells derived from lots FCL and DUX at P7 (*Figure 6B*). hCLiPs derived from lot JFC were contaminated by an increased percentage of fibroblast-like cells; therefore, we karyotyped FACS-sorted EPCAM$^+$ cells (at the first passage) which were then passaged four times after sorting (*Figure 6B*). None of the analyzed cells exhibited any chromosomal abnormality (20 cells analyzed per lot) and all the analyzed cells were diploid (50 cells analyzed per lot) (*Figure 6B*). This implies that hCLiPs were derived from diploid hepatocytes, which is consistent with our previous observations in rat CLiPs (*Katsuda et al., 2017*). We further investigated transcriptomic changes in hCLiPs derived from lots FCL and DUX between P0 and P10 using cells from the experimental batches that maintained higher levels of EPCAM and CD24 expression (*Figure 6—figure supplement 1B*) (experimental batch #3 and #2 for lots DUX and FCL, respectively). A heatmap of genes that were differentially expressed between P0 and P10 showed that the phenotype of hCLiPs gradually changed (*Figure 6—figure supplement 1C*). As indicated on the right in *Figure 6—figure supplement 1C*, genes whose expression decreased included those related to hepatic functions, indicating that hCLiPs lost their hepatic phenotypes during repeated passage. Nonetheless, the heatmap suggested that hCLiPs retained at least some of their original characteristics until approximately P5 (*Figure 6—figure supplement 1C*). Thus, we investigated the hepatic phenotype of hCLiPs at P3 and P5. qRT-PCR analysis of hCLiPs derived from each lot indicated that absolute expression levels of hepatic genes consistently decreased as the passage number increased (*Figure 6C*). Nevertheless, hCLiPs derived from each lot, particularly lots FCL and DUX, could undergo hepatic differentiation (*Figure 6C*). Immunocytochemistry revealed that Hep-i (+) cells derived from lot FCL expressed hepatic marker proteins at P3 (*Figure 6—figure supplement 1D*). We also investigated CYP enzymatic activities in these cells. Although the CYP enzymatic activities clearly decreased upon repeated passage, the basal activities of these enzymes, with the exception of CYP2C19, were maintained at P3 and P5 (*Figure 6D*). Induction of CYP3A enzymatic activity in response to rifampicin and phenobarbital was relatively stable even at P3 and P5, especially in Hep-i(+) cells (*Figure 6E*). In summary, functional decline of hCLiP-derived hepatocytes during continuous culture is unavoidable; however, CYP3A, the most important CYP in human drug metabolism, is still induced in these cells.

## Enrichment of hCLiPs with LPC markers and characterization of their descendants in the subsequent culture

We then asked whether the loss of the original phenotype of hCLiPs, especially the hepatic phenotype, during serial passages would be caused by their own phenotypic change or by expansion of contaminated NPCs. Using antibodies against three LPC makers (EPCAM, PROM1 and CD24) and a NPC marker THY1/CD90, which particularly characterizes fibroblastic cells, we sorted each LPC-marker$^+$THY1$^-$ population and LPC-marker$^-$THY1$^+$ population from FCL-hCLiPs at P0 (*Figure 7A*, *Figure 7—figure supplement 1*). qRT-PCR clearly demonstrated that each LPC marker enabled enrichment of cells with hepatic phenotype (*ALB*, *TTR*, *GJB1*), whereas THY1-enriched cells consistently exhibited mesenchymal phenotype as characterized by the expression of *ACTA2* and *VIM* in addition to *THY1* (*Figure 7B*). After subsequent three passages (2–3 weeks), LPC marker-enriched cells

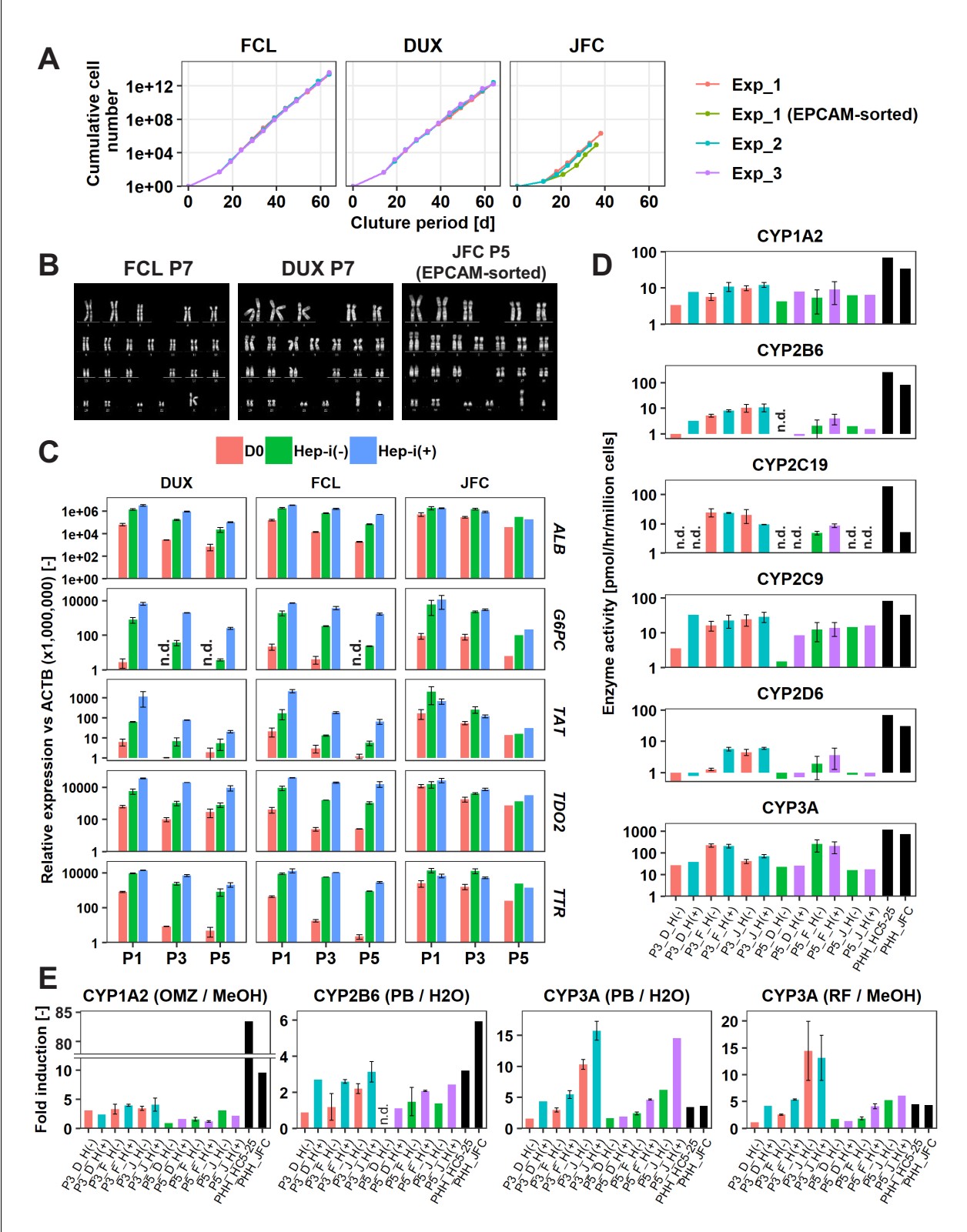

**Figure 6.** hCLiPs stably expand in vitro and retain their hepatic differentiation ability. (**A**) Growth curves of hCLiPs from P0–10 (lots FCL and DUX) or P0–four or P0–5 (lot JFC). Each curve represents data obtained in independent experiments. Data in each plot indicate the cumulative cell numbers at each time point normalized against that at D0 (set to one cell). (**B**) Representative chromosomal images of hCLiPs derived from the three lots, as assessed by Q-band karyotyping. (**C**) qRT-PCR analysis of hepatocyte-specific genes at P1, P3, and P5. Data are normalized against *ACTB* expression, and shown as

*Figure 6 continued on next page*

*Figure 6 continued*

mean ± SEM of two repeated experiments except JFC cells at P5 (n = 1). (D) Basal enzymatic activities of major CYPs in Hep-i(-) and Hep-i(+) cells at P3 and P5, as well as in PHHs, as assessed by LC-MS/MS using a cocktail of substrates. Data are shown as one experiment or the mean ± SEM of two repeated experiments for each lot of hCLiPs and the results of one experiment for each lot of PHHs. N.d. indicates 'not detected'. (E) Inducibility of CYP1A2, CYP2B6, and CYP3A activities at P3 and P5. Enzymatic activities in inducer-treated cells were compared with those in cells treated with the corresponding vehicle by LC-MS/MS analysis using a cocktail of substrates. Data are shown as one experiment or the mean ± SEM of two repeated experiments for each lot of hCLiPs and the results of one experiment for each lot of PHHs. N.d. indicates 'not detected'.

The online version of this article includes the following figure supplement(s) for figure 6:

**Figure supplement 1.** Characterization of hCLiPs upon long-term culture.

relatively retained their LPC/hepatic phenotypes as assessed by qRT-PCR (*Figure 7C*) and flow cytometry (*Figure 7—figure supplement 1*) compared to THY1-enriched cells. However, we noted that compared to the cells at P0, such LPC/hepatic phenotypes were largely reduced after three passages (*Figure 7—figure supplement 2*). These results demonstrated that, even after enrichment of LPC marker+ cells, phenotypic deterioration of hCLiPs is unavoidable. Since hCLiPs did not exhibit remarkable morphological changes until P10 of culture (*Figure 6—figure supplement 1A*), the results obtained here call attention to the need for a careful quality control of hCLiPs by quantitative analyses, including flow cytometry and qRT-PCR.

## Repopulation of chronically injured mouse livers by hCLiPs

The capacity to repopulate injured livers is the most important and stringent criterion of a candidate cell source for liver regenerative medicine. Depending on the disease, 1–15% of hepatocytes must be replaced to achieve and sustain a therapeutic effect (*Jorns et al., 2012*; *Rezvani et al., 2016*). Laboratory-generated hepatocytes typically have RIs of less than 5% (*Rezvani et al., 2016*), but a few studies reported maximum RIs of 20% or 30% in individual animals (*Carpentier et al., 2014*; *Du et al., 2014*). Moreover, in a recent study, *Zhang et al. (2018)* achieved much higher RI (>60%) by transplanting expandable hepatic cells named ProliHH, which were generated from PHHs as with hCLiPs. Thus, it is important to evaluate the repopulative capacity of hCLiPs from a comparative point of view.

We assessed the repopulative capacity of hCLiPs in immunodeficient mice with chronically injured livers. Our previous study revealed that rat CLiPs repopulate the liver of cDNA-uPA/SCID mice (*Katsuda et al., 2017*); therefore, we first transplanted hCLiPs derived from lots FCL, DUX and JFC at P0–P2 into this model. After intrasplenic transplantation of primary hCLiPs that had been expanded in vitro for approximately 2 weeks (11–13 days) (hereafter designated P0-hCLiPs), the human ALB (hALB) level was exponentially increased in the blood of some, but not all, mice (*Figure 8A*, red lines). The maximum hALB level in blood was >10 mg/ml, which is comparable with that observed following transplantation of PHHs in this animal model (*Tateno et al., 2015*). Immuno-histochemistry (IHC) of human-specific CYP2Cs (including CYP2C9 and other CYP2Cs according to the manufacturer's datasheet) demonstrated extensive repopulation in mouse livers extracted at 10–11 weeks after transplantation (*Figure 8B*). Although the RI varied among mice (32.2 ± 13.5% for lot FCL, n = 11; 39.3 ± 13.5% for lot JFC, n = 11; 17.8 ± 16.4% for lot DUX, n = 4, mean ± SEM), it reached >90% in some animals (*Figure 8C*). This maximum RI is comparable with that achieved after transplantation of PHHs (*Rezvani et al., 2016*). The repopulative capacity declined as the culture period increased (*Figure 8A and C*). Nonetheless, one mouse transplanted with FCL-P1-hCLiPs (hCLiPs derived from lot FCL that were passaged once before transplantation) (67.4%) and two mice transplanted with JFC-P2-hCLiPs (hCLiPs derived from lot JFC that were passaged twice before transplantation) (83.1% and 91.1%) exhibited high RIs. It should be noted that FCL-P1-hCLiPs and JFC-P2-hCLiPs underwent approximately 1000- and 400-fold expansion from the initial PHHs, respectively. These fold expansion is comparable to that achieved by ProliHH developed by *Zhang et al. (2018)*. They reported high repopulative capacity of ProliHH at P4-P6. The fold expansion based on the initial number of PHHs, ProliHH at P4-P6 in their culture system underwent approximately 400–1000-fold expansion. Thus, hCLiPs have repopulative capacity at the comparable levels with ProliHH. We further confirmed the repopulative capacity of FCL-P0-hCLiPs using another model, namely, TK-NOG mice (*Hasegawa et al., 2011*). In this model, the serum hALB level was

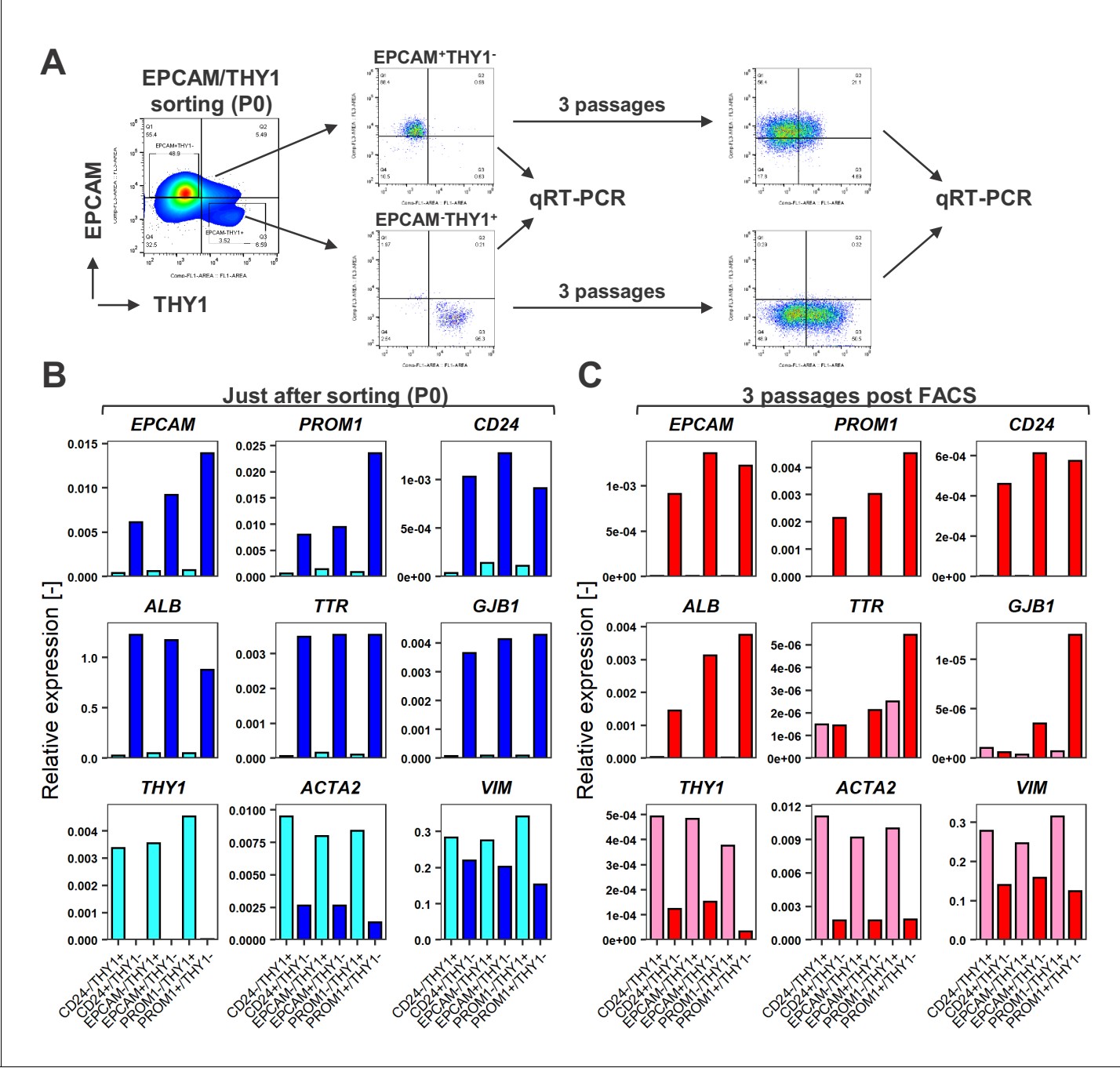

**Figure 7.** FACS is useful for enrichment of hCLiPs, but does not prevent phenotypic deterioration of the descendant cells in the subsequent culture. (**A**) Schematic representation of the experimental design for FACS of LPC marker[+]THY1[-] cells (hCLiP-enriched cells) and LPC marker[-]THY1[+] cells (putative NPCs), and the subsequent evaluation by qRT-PCR. Results for the EPCAM/THY sorting experiments are partially shown as a representative of this study. The full result for EPCAM/THY1, PROM1/THY1 and CD24/THY1 sorting experiments are shown in *Figure 5—figure supplement 1*. (**B**) qRT-PCR was performed for the cells just after sorting using LPC markers (*EPCAM*, *PROM1* and *CD24*), hepatic markers (*ALB, TTR* and *GJB1*) and fibroblast markers (*THY1, ACTA1* and *VIM*). (**C**) qRT-PCR using the same gene panel as in (**B**) was performed for the sorted cells which were plated and subsequently cultured in the standard hCLiP culture condition for another 2 weeks. For comparison between P0 (as shown in B) and these descendant cells are shown in *Figure 7—figure supplement 2*.

The online version of this article includes the following figure supplement(s) for figure 7:

**Figure supplement 1.** Detailed data for FACS of FCL-hCLiPs using antibodies for LPC markers and THY1.

**Figure supplement 2.** Comparison of gene expression levels of FCL-hCLiPs just after sorting and their descendants which underwent another 2 week culture.

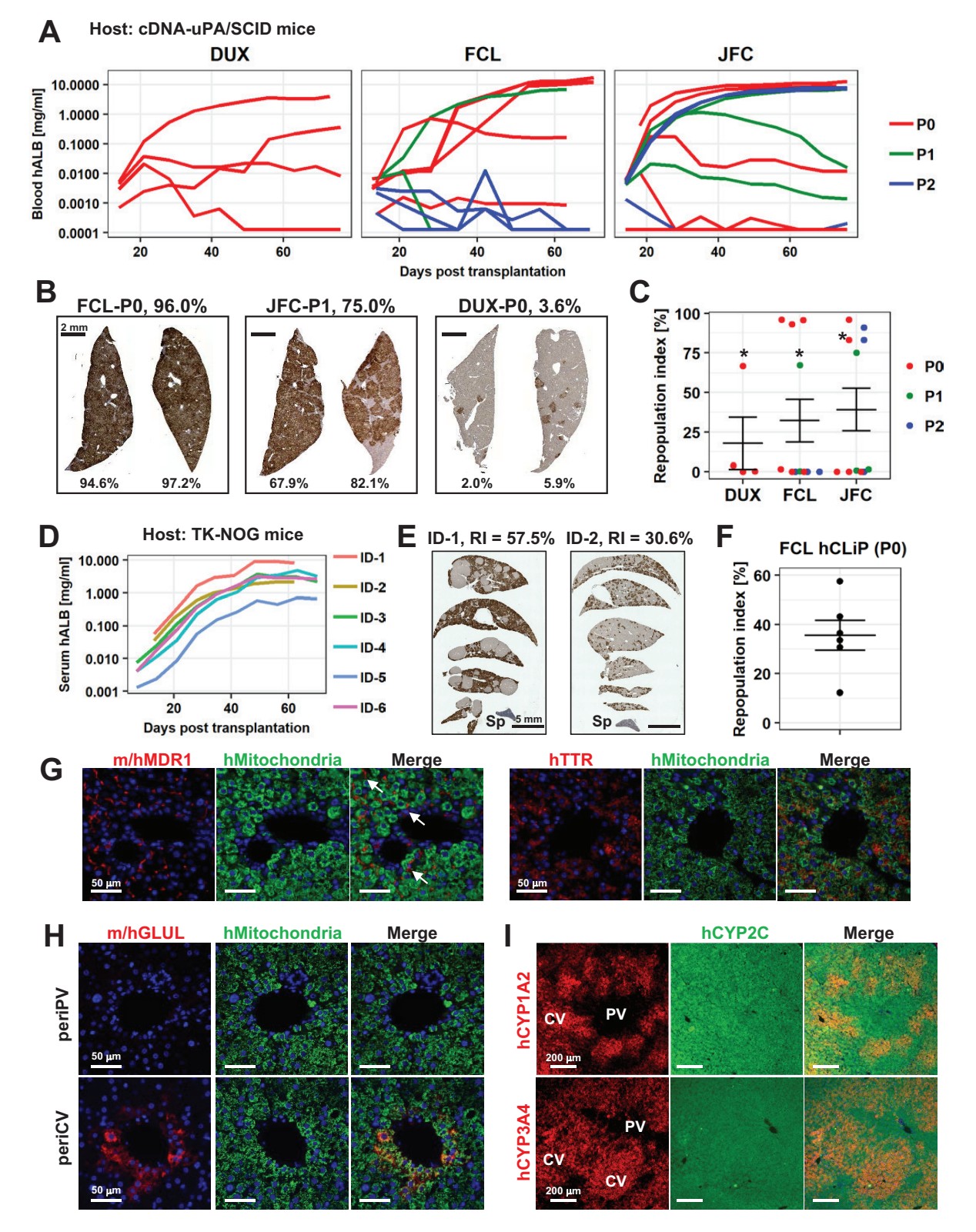

**Figure 8.** hCLiPs repopulate chronically injured mouse livers and contribute to reconstruction of the normal liver architecture. (A) hALB levels in blood of cDNA-uPA/SCID mice. Each line indicates the level in an individual mouse. Colors denote the passage number of transplanted hCLiPs. (B) Representative images of cDNA-uPA/SCID mouse livers highly (left and middle panels) and slightly (right panel) repopulated by hCLiPs. The percentages indicate RIs. (C) Distribution of RIs in livers of cDNA-uPA/SCID mice at 10–11 weeks after transplantation of hCLiPs, as assessed by IHC of

*Figure 8 continued on next page*

*Figure 8 continued*

CYP2C (shown in B). Colors denote the passage number of transplanted hCLiPs. RIs were calculated for samples marked by asterisks using hepatocytes isolated from chimeric livers by two-step collagenase perfusion followed by incubation with magnetic beads conjugated with a specific anti-mouse antibody (see Materials and methods for details). Bars indicate the mean ± SEM. (D) hALB levels in sera of TK-NOG mice. Each line indicates the level in an individual mouse. (E) Representative IHC of human CYP2C in TK-NOG mouse livers highly (left panel) and intermediately (right panel) repopulated by hCLiPs. The percentages indicate RIs determined based on this IHC. (F) A dot plot showing the distribution of RIs in livers of cDNA-uPA/SCID mice at 10–11 weeks after transplantation of hCLiPs. Bars indicate the mean ± SEM. (G) IHC of the hepatic function marker proteins MDR1 (left panels) and TTR (right panels). Sections were counterstained with an anti-human mitochondria antibody (green) and DAPI. Images of sections transplanted with hCLiPs derived from lot FCL are shown as representative data. (H) IHC of the zone 3-specific protein GLUL. Sections were counterstained with an anti-human mitochondria antibody (green) and DAPI. Images of sections transplanted with hCLiPs derived from lot FCL are shown as representative data. (I) IHC of the zone 3-specific CYPs CYP1A2 and CYP3A4. Sections were counterstained with an antibody against human CYP2C, which does not show strong zone specificity. Nuclei were also counterstained with DAPI in merged images. Images of sections transplanted with hCLiPs derived from lot FCL are shown as representative data. PV and CV indicate portal vein and central vein, respectively.

The online version of this article includes the following figure supplement(s) for figure 8:

**Figure supplement 1.** Repopulation assay of FCL-hCLiPs at later passages.

dramatically elevated to at most 8.1 mg/ml (*Figure 8D*). The maximum RI was lower in TK-NOG mice (57.5%) than in cDNA-uPA/SCID mice (96.0%) (*Figure 8E and F*). However, engraftment was more efficient in TK-NOG mice than in cDNA-uPA/SCID mice; significant repopulation (>15% RI) with FCL-P0-hCLiPs was observed in 83% (5/6 mice) of TK-NOG mice (*Figure 8F*), but only in 50% (3/6 mice) of cDNA-uPA/SCID mice (*Figure 8C*). Examination of the area repopulated by hCLiPs by staining with an antibody against human mitochondria showed that repopulating human cells expressed MDR1 and TTR, which are associated with hepatic function (*Figure 8G and H*). MDR1 was detected on the apical side of adjacent mouse and human hepatocytes, suggesting that hCLiP-derived cells successfully reconstructed the normal liver architecture (*Figure 8G*, arrows). Accordingly, hepatic zonation was correctly established in the repopulated regions, as assessed by expression of glutamate-ammonia ligase (GLUL, also known as glutathione synthetase) (*Figure 8H*), CYP1A2 and CYP3A4 (*Figure 8I*).

We next tested whether, after several passages, hCLiPs would still retain repopulative capacity. First, we transplanted FCL-P4-hCLiPs (hCLiPs derived from lot FCL that were passaged four times before transplantation) to cDNA-uPA/SCID mice. Based on the growth curves (*Figure 6A*), we estimated that these cells underwent $2.2 \pm 0.94 \times 10^6$ fold (n = 3, mean ± SEM) expansion from the initial PHHs. We observed increase of blood hALB levels in these mice, but the hALB increasing rates were much slower than in the animals transplanted with FCL-P0-hCLiP or FCL-P1-hCLiPs (*Figure 8—figure supplement 1A*). The hALB levels 8 weeks after transplantation was 10 μg/ml at most (*Figure 8—figure supplement 1A*), which was approximately 1/1000 of the mice with high RI (>60% RI). As expected, hCYP2C staining indicated that all the mice transplanted with FCL-P4-hCLiPs exhibited very low RI (<1%) (*Figure 8—figure supplement 1B*). We also transplanted FCL-P3-hCLiPs to TK-NOG mice. In this experiment, we prepared EPCAM-expressing cells at the first passage by magnetic activated cell sorting (MACS), and cultured them for another two passages (three passages in total). During the culture at P3, we separated these cells to two groups, one with hepatic induction (P3 Hep-i(+)) and the other without hepatic induction (P3 Hep-i(-)). After transplantation to TK-NOG mice, we observed serum hALB increase in these mice (*Figure 8—figure supplement 1C*). Importantly, we confirmed that Hep-i(+) cell-transplanted group showed consistently higher hALB levels than the Hep-i(-) cell-transplanted group. However, the serum hALB levels of FCL-P3-hCLiP-Hep-i(+)-transplanted mice at 8 weeks were still only 9.1 ± 1.8 μg/ml (n = 4, mean ± SEM), which were again much lower than the mice transplanted with FCL-P0-hCLiPs. These results collectively show that hCLiPs unavoidably decrease their repopulative capacity following extended in vitro culture.

## Functional characterization of hCLiP-derived hepatocytes in chimeric livers

Finally, we isolated human cells from chimeric mouse livers and investigated their functionality because it has been argued that some types of laboratory-generated hepatocytes are not fully functional after repopulation (*Rezvani et al., 2016*). We first performed microarray-based transcriptomic analysis. After isolating hepatocytes from chimeric livers of cDNA-uPA/SCID mice by a two-step

collagenase perfusion method, we eliminated mouse cells using a magnetic bead separation system. Microscopic observation revealed that 32.7%, 16.8% and 33.1% of hepatocytes isolated from chimeric livers of mice transplanted with hCLiPs derived from lots FCL, JFC and DUX bound to magnetic beads conjugated with a specific anti-mouse antibody prior to magnetic separation, respectively, while these percentages were reduced to 2.9%, 0.0%, and 1.6% after magnetic separation, respectively. Thus, we assumed that the results of experiments performed with these cells should be mostly ascribed to human cells. Magnetically separated human cells exhibited typical morphologies of mature hepatocytes (*Figure 9A*). However, unexpectedly, hierarchical clustering and principle component analysis (PCA) of the entire transcriptome showed that chimeric liver-derived human cells were distinct from PHHs (*Figure 9B and C*). A control sample of human hepatocytes isolated from chimeric livers following transplantation of IPHHs (lot JFC) yielded similar results as human hepatocytes isolated from chimeric livers following transplantation of hCLiPs (*Figure 9B and C*), indicating that the transcriptomic difference between human hepatocytes in chimeric livers and PHHs is due to environmental differences between human and mouse livers. Surprisingly, GSEA demonstrated that multiple hepatic function-related gene sets were overrepresented in human hepatocytes isolated from chimeric livers in comparison with PHHs (*Supplementary file 6*). The majority of these gene sets were associated with metabolic pathways. Other hepatic functions were also enriched, such as pathways associated with coagulation and complement production (*Figure 9D*, *Supplementary file 6*). BEC/LPC marker genes were underrepresented in hCLiP-derived hepatocytes isolated from chimeric livers and PHHs in comparison with hCLiPs (*Figure 9—figure supplement 1A*), demonstrating that hCLiPs underwent hepatic maturation after repopulating mouse livers. We also investigated whether hCLiP-derived hepatocytes isolated from chimeric livers exhibited CYP activities. As expected based on the transcriptomic analysis, hCLiP-derived cells isolated from chimeric livers exhibited basal enzymatic activities of major CYPs at the levels comparable with those in PHHs (*Figure 9E*). Enzymatic activities of CYP1A2, CYP2B6 and CYP3A were markedly induced in hCLiP-derived hepatocytes isolated from chimeric livers upon treatment with rifampicin, phenobarbital and omeprazole (*Figure 9F*). Consistently, qRT-PCR analysis demonstrated that expression of *CYP1A2*, *CYP2B6* and *CYP3A4* was dramatically upregulated upon treatment with CYP inducers (*Figure 9—figure supplement 1B*). Finally, activities of the phase II enzymes SULT and UGT in hCLiP-derived hepatocytes isolated from chimeric livers were comparable with those in PHHs (*Figure 9G*). These results indicate that although their transcriptomic profiles are not identical to those of PHHs, including IPHHs and APHHs, hCLiPs functionally mature in the mouse liver.

## Discussion

In this study, we demonstrated that hCLiPs can repopulate chronically injured livers of immunodeficient mice. An efficient repopulative capacity is one of the most important requirements of a candidate cell source for transplantation therapy; however, it is very challenging to develop such a cultured cell source. Laboratory-generated hepatic cells, such as pluripotent cell-derived hepatic cells and those transdifferentiated from cells of different lineage origins, have a poor repopulative capacity (*Rezvani et al., 2016*). The RI of laboratory-generated hepatocytes is typically less than 5% (*Rezvani et al., 2016*). After our report of rodent CLiPs (*Katsuda et al., 2017*), four groups recently reported methods for in vitro generation of proliferative liver (progenitor) cells from human hepatocytes (*Fu et al., 2018*; *Hu et al., 2018*; ; *Kim et al., 2019*; *Zhang et al., 2018*). In three of these studies (*Fu et al., 2018*; *Hu et al., 2018*; ; *Kim et al., 2019*), the generated cells exhibited relatively low repopulative efficiency with approximately 13% of RI at maximum. In contrast, Zhang el al. reported strikingly high repopulation efficiency with as high as 64% of RI (*Zhang et al., 2018*). Importantly, although the proliferative efficiency is limited compared with IPHHs, they succeeded in induction of proliferative hepatic cells even from APHHs. Moreover, contrary to the dichotomous repopulation of hCLiPs in our study (RI >80% or nearly 0%), Zhang et al demonstrated highly stable repopulation among transplanted animals. Our study is, thus, not the first one to report substantial repopulation using an in vitro-generated human hepatic cell source. Nonetheless, to solidify a novel concept, more evidence must be provided independently from multiple laboratories. As such, we still believe that our work also plays an important role in pioneering this new field.

Another important finding in this study is that hCLiPs may be a novel cell source for drug discovery studies. The major criterion for the application of cultured hepatic cells in drug discovery studies,

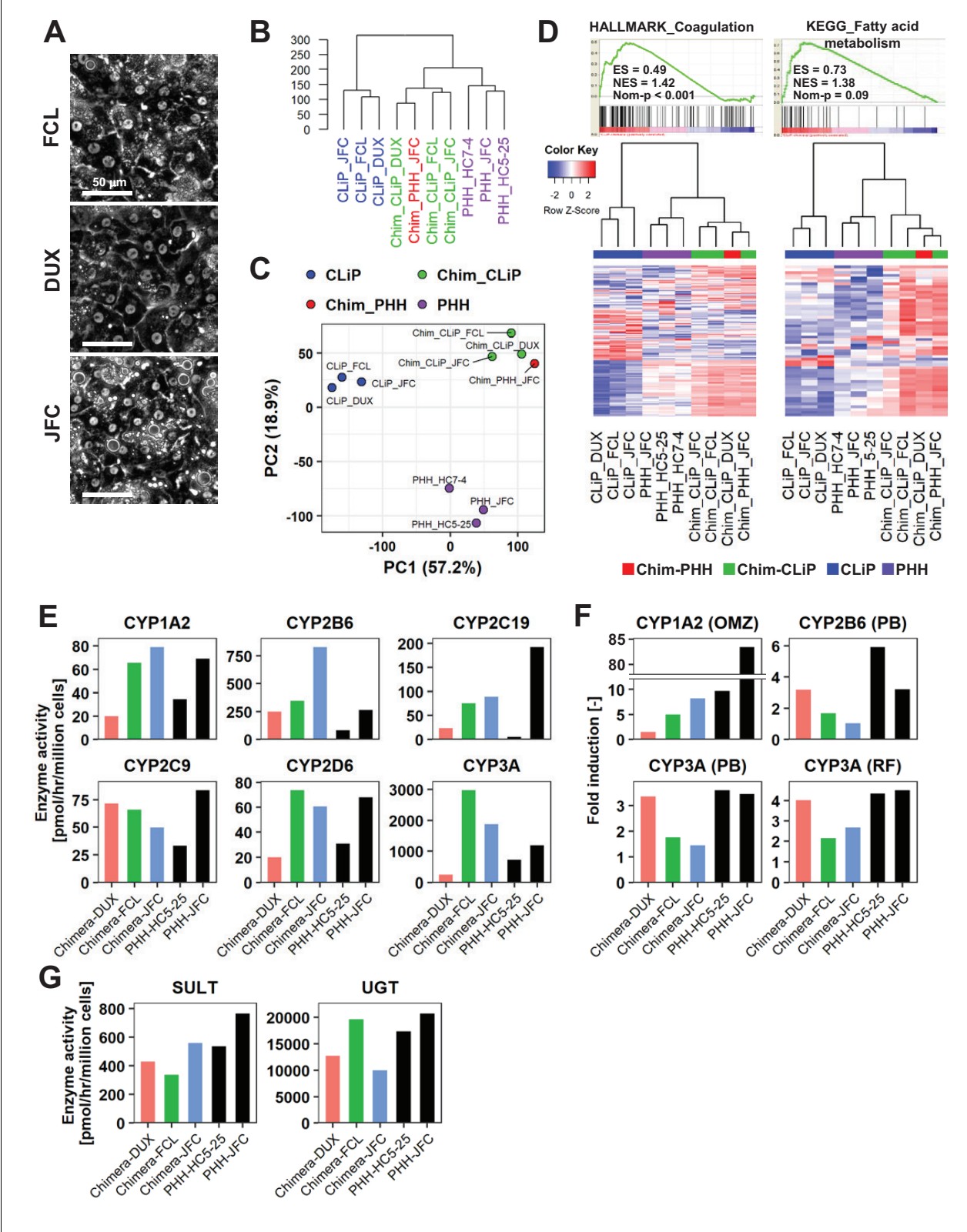

**Figure 9.** Human cells isolated from chimeric livers of mice transplanted with hCLiPs have mature functions. (A) Phase contrast images of human cells isolated from chimeric livers of mice transplanted with hCLiPs. (B) Hierarchical clustering based on Euclidean distance of the entire transcriptome (27,459 probes) comparing hCLiPs prior to transplantation (hCLiP), hCLiP-derived hepatocytes from chimeric livers (transplanted cells were at P1, P0, and P0 for lots FCL, DUX, and JFC, respectively), and PHHs. Data for human hepatocytes isolated from chimeric livers of mice transplanted with PHHs

*Figure 9 continued on next page*

*Figure 9 continued*

(lot JFC) are shown for reference. (C) PCA mapping of the samples described in (B). (D) Gene sets enriched in hCLiP-derived cells from chimeric livers in comparison with PHHs (top panels) and their corresponding heatmaps (bottom panels). Hierarchical clustering was performed based on Euclidean distance. (E) Basal enzymatic activities of major CYPs in hCLiP-derived cells from chimeric livers and PHHs, as assessed by LC-MS/MS using a cocktail of substrates. Each value is determined by one experiment with two replicate cultures. (F) Inducibility of CYP1A2, CYP2B6, and CYP3A activities. Enzymatic activities in inducer-treated cells were compared with those in cells treated with the corresponding vehicle by LC-MS/MS analysis using a cocktail of substrates. Each value is determined by one experiment with two replicate cultures. (G) Activities of the phase II enzymes UGT and SULT, as assessed by LC-MS/MS analysis using a cocktail of substrates. Each value is determined by one experiment with two replicate cultures.

The online version of this article includes the following figure supplement(s) for figure 9:

**Figure supplement 1.** Characterization of human cells isolated from chimeric livers of mice transplanted with hCLiPs.

particularly to evaluate the functions of drug-metabolizing enzymes, is the inducibility of CYP enzymatic activities. CYP enzymes play central roles in the metabolism of clinically used drugs and xenobiotics. In general, CYP induction accelerates the clearance of xenobiotics, leading to beneficial or harmful outcomes depending on the context. Thus, recapitulation of CYP induction in cultured hepatocytes or their equivalents is important to precisely predict the effects of a tested drug on hepatocytes. However, PHHs lose their hepatic functions, including CYP inducibility, upon in vitro culture. Laboratory-generated hepatocytes reportedly exhibit basal CYP activities after maturation (*Baxter et al., 2015*; *Kanninen et al., 2016*; *Liu et al., 2011*; *Takayama et al., 2018*; *Takayama et al., 2014*). Although a few groups described CYP inducibility in terms of enzymatic activity (*Inamura et al., 2011*; *Pettinato et al., 2016*; *Takayama et al., 2012*), such reports are very limited, to the best of our knowledge. We propose that hCLiPs are a novel platform for drug discovery studies.

An issue yet to be addressed is clarification of the mechanism underlying the small molecule-mediated conversion of PHHs to hCLiPs. Mini-screen of three small molecules Y, A and C demonstrated that A and C individually accelerated proliferation of PHHs, while Y alone exhibited no beneficial effect on proliferation, and even negatively affected proliferation when combined with AC (in comparison of YAC with AC). This is in line with our previous observation in rodent hepatocyte culture, in which Y minimally affected the proliferation of these three small molecules (*Katsuda et al., 2017*). AC substantially induced the proliferation of rodent hepatocytes at the comparable, if not at the same, level with YAC (*Katsuda et al., 2017*). Thus, the synergistic effect of A83-01 and CHIR99021 is the key to hepatocyte proliferation in rodent and human hepatocytes. Importantly, comparative analysis of APHHs and IPHHs suggested that activity of Wnt signaling in response to CHIR99021 may partly explain the proliferative ability of IPHHs. On the other hand, APHHs responded to A83-01 equally or even more efficiently than IPHHs, leaving a question how A83-01 affected the proliferation of IPHHs, but not APHHs. Since A83-01 is essential to IPHH proliferation as assessed in the mini-screen (*Figure 1A*), this small molecule might affect IPHHs in a TGFβ-independent manner. Further investigation is needed to fully understand the difference between the proliferative ability endowed by FAC between IPHHs and APHHs. Another important issue to be considered is the requirement for FBS in hCLiP induction, which is not the case for rodent CLiP induction. FBS-derived factor(s) essential for hCLiP induction should be identified in a future study.

Comparison of our study with the recently reported four studies provides hints to mechanistic understanding of in vitro PHH expansion (*Fu et al., 2018*; *Hu et al., 2018*; *Kim et al., 2019*; *Zhang et al., 2018*). Notably, hepatocyte growth factor (HGF), which is not included in our culture condition, is used in all these four studies, suggesting its critical role. Indeed, *Kim et al. (2019)* particularly emphasizes its essential role in the presence of AC. On the other hand, *Zhang et al. (2018)* ascribe the proliferative capacity of PHHs particularly to Wnt signaling (*Zhang et al., 2018*). Interestingly, these authors reported that Wnt3a plays an essential role, while neither CHIR99021 nor Wnt signaling amplifier Rspo1 substituted for the pro-proliferative effect of Wnt3a. Moreover, these authors proposed a unique idea that hypoxic culture condition supports the stable proliferation of PHHs by suppressing PHH senescence. In line with this observation, Fu et al. also demonstrated that a sirtuin suppressor nicotinamide decreases proliferation of PHH. This finding highlights the difference between human and rodent PHHs: nicotinamide is known to induce proliferation of rat hepatocytes (*Mitaka et al., 1991*) and thus is frequently added to hepatocyte culture medium (including

ours), but this may not be the case for induction of PHH proliferation. Collectively, these findings, including ours, provide important insight to optimization of the methodology of PHH expansion.

## Materials and methods

### Primary human hepatocytes

Infant primary human hepatocytes (IPHHs) (lots FCL, DUX, JFC and MRW) were purchased from Veritas Corporation (Tokyo, Japan). Adult primary human hepatocytes (APHHs) (lots HC1-14, HC3-14, HC5-25, and HC7-4) were purchased from Sekisui XenoTech (KS). IPHH lot 187273 and APHH lot 187271 were purchased from Biopredic (Saint-Gregoire, France). Donor information is summarized in *Table 1*.

### Culture medium

The basal medium for culture of PHHs was SHM (DMEM/F12 (Life Technologies, MA) containing 2.4 g/l $NaHCO_3$ and L-glutamine) (*Chen et al., 2007*; *Katsuda et al., 2018*) supplemented with 5 mM HEPES (Sigma, MO), 30 mg/l L-proline (Sigma), 0.05% bovine serum albumin (Sigma), 10 ng/ml epidermal growth factor (Sigma), insulin-transferrin-serine-X (Life Technologies), $10^{-7}$ M dexamethasone (Sigma), 10 mM nicotinamide (Sigma), 1 mM ascorbic acid-2 phosphate (Wako, Osaka, Japan), and antibiotic/antimycotic solution (Life Technologies). Depending on the experiment, this basal medium was supplemented with 10% FBS (Life Technologies), as well as small molecules, namely, 10 μM Y-27632 (Wako), 0.5 μM A-83–01 (Wako), and 3 μM CHIR99021 (Axon Medchem, Reston, VA). After a mini-screen of these three small molecules, PHHs were routinely cultured in SHM supplemented with 10% FBS, 0.5 μM A-83–01, and 3 μM CHIR99021.

### Induction of hCLiPs from IPHHs

IPHHs were thawed in a water bath set to 37°C and suspended in 10 ml Leibovitz's L-15 Medium (Life Technologies) supplemented with Glutamax (Life Technologies) and antibiotic/antimycotic solution. After centrifugation at 50 × g for 5 min, the cells were resuspended in William's E medium supplemented with 10% FBS, GlutaMAX, antibiotic/antimycotic solution, and $10^{-7}$ M insulin (Sigma). The number of viable cells was determined using trypan blue (Life Technologies). IPHHs from lot JFC were seeded in collagen I-coated plates (IWAKI, Shizuoka, Japan) at a density of approximately $5 × 10^3$ viable cells/$cm^2$. IPHHs from lots FCL and DUX barely attached to the plates, and many of the small number that did attach subsequently detached prior to D3, as monitored by time-lapse imaging using a BZ-X700 microscope (Keyence, Osaka Japan) (data not shown). Therefore, IPPHs from lots FCL and DUX were seeded at a density of approximately $2 × 10^4$ viable cells/$cm^2$, which was approximately 4-fold higher than the seeding density of IPHHs from lot JFC. To determine the fold change in cell number during in vitro culture, the number of adherent cells on D3 was counted based on micrographs acquired at 10 × magnification (5–10 fields per experiment).

### Subculture of hCLiPs

Cells were harvested using TrypLE Express (Life Technologies, MA) when they reached 70–100% confluency and then re-plated into a 10 cm collagen-coated plate at a density of $1–2 × 10^5$ cells/dish.

### Cell proliferation assay

Numbers of viable cells were estimated based on the WST-8 assay using Cell Counting Kit 8 (Dojindo, Kumamoto, Japan), according to the manufacturer's instructions.

### Flow cytometry and cell sorting

Flow cytometry and cell sorting were performed using a S3e Cell Sorter (BioRad, Hercules, CA). Cells were labeled with APC-conjugated mouse anti-human CD44 (1:20; G44-26; BD, Franklin Lakes, NJ), APC-conjugated mouse anti-human EPCAM (1:20; EBA-1; BD), PE/Cy7-conjugated anti-human/mouse CD49f (ITGA6) (1:20; GoH3; Biolegend), PE/Cy7-conjugated anti-human CD24 (1:20; ML5; Biolegend), APC-conjugated human anti-PROM1/CD133 (1:11; AC133; Miltenyi Biotech), APC-conjugated mouse anti-human CD26/DPP4 (1:11, FR10-11G9; Miltenyi Biotech), and FITC-conjugated

mouse anti-human CD90/THY1 antibodies. An APC-conjugated mouse IgG1, κ isotype control antibody (Biolegend, MOPC-21) and a PE-Cy7-conjugated mouse IgG2b, κ isotype control (BD, 27–35) were used as controls.

## Flow cytometry of primary mouse hepatocytes

An adult Rosa$^{YFP/YFP}$ mouse received retro-orbital injection of AAV-TBG-cre at the dose of 2.5 $\times$ 10$^{11}$ viral particles, and hepatocytes were harvested 3 weeks later using a standard two-step collagenase perfusion method. Isolated Rosa$^{YFP/YFP}$ hepatocytes were stained with APC anti-mouse CD326/EPCAM) (1:100, G8.8, Biolegend) or APC anti-human/mouse CD49f/Itga6 antibody (1/100, GoH3, Biolegend). For staining with Prom1/Cd133 and Cd24, cells were incubated with purified rat anti-mouse CD133/Prom1 antibody (1:100, 315–2 C11, Biolgend) and rat anti-mouse CD24 antibody (1:100, Biolegend, M1/69) followed by staining with Alexa647-conjugated donkey anti-rat antibody (1:300, Jackson ImmunoResearch). APC-conjugated rat IgG2a or IgG2b, κ isotype control antibody was used as control. DAPI was added to stain dead cells. Attune NxT Flow Cytometer (Lifetechnologies) was used for data collection. B6 wild-type adult hepatocytes, which were stained with only DAPI, was used for making the threshold of YFP signal.

## Microarray analysis

One-color microarray-based gene expression analysis was performed using a SurePrint G3 Human Gene Expression v3 8 $\times$ 60K Microarray Kit (Agilent, Santa Clara, CA) following the manufacturer's instructions. The 75th percentile shift normalization was performed using GeneSpring software (Agilent).

## Induction of hepatic differentiation of hCLiPs

hCLiPs were harvested using TrypLE Express (Life Technologies) and reseeded into a collagen I-coated 24-well plate at a density of 5 $\times$ 10$^4$ cells/well (2.5 $\times$ 10$^4$ cells/cm$^2$). When cells reached approximately 50–80% confluency, culture medium was replaced by SHM supplemented with 2% FBS, 0.5 mM A-83–01, and 3 mM CHIR99021 in the absence (Hep-i(-)) or presence (Hep-i(+)) of 5 ng/ml human OSM (R and D) and 10$^{-6}$ M dexamethasone. Cells were cultured for a further 6 days and fresh medium was provided every 2 days. On D6, cells were overlaid with a mixture of Matrigel (Corning, Corning, NY) and the aforementioned hepatic induction medium at a ratio of 1:7 and cultured for another 2 days. Thereafter, Matrigel was removed via aspiration, samples were washed with Hank's Balanced Salt Solution supplemented with Ca$^{2+}$ and Mg$^{2+}$ (Life Technologies), and cells were used for RNA extraction or CYP induction experiments.

## CYP induction

SHM containing 2% FBS, but not A-83–01 or CHIR99021, was used as basal medium. CYP3A and CYP2B6 were induced via treatment with 10 µM rifampicin and 1 mM phenobarbital. An equal volume of methanol (1/100 dilution) and H$_2$O (1/1000 dilution) was used as the vehicle control for rifampicin and phenobarbital, respectively. CYP1A2 was induced via treatment with 50 µM omeprazole, and methanol (1/100 dilution) was used as the vehicle control. Each CYP induction medium was replaced by freshly prepared medium every day. After 3 days, CYP activity was measured by LC-MS/MS.

## CYP activity assay using a cocktail of substrates

Cells were cultured in phenol red-free William's E medium supplemented with a cocktail of substrates (1/100 dilution) at 37°C for 1 hr. This cocktail contained 40 µM phenacetin as a CYP1A2 substrate, 50 µM bupropion as a CYP2B6 substrate, 0.1 µM amodiaquin as a CYP2C8 substrate, 5 µM diclofenac as a CYP2C9 substrate, 100 µM S-mephenytoin as a CYP2C19 substrate, 5 µM bufuralol as a CYP2D6 substrate, 5 µM midazolam as a CYP3A substrate, and 100 µM 7-hydroxycoumarin as a UGT and SULT substrate. Thereafter, the culture supernatant was harvested and metabolites were quantified by LC-MS/MS as described previously (*Kozakai et al., 2012*) with minor modifications.

## Measurement of CYP protein expression

CYP protein levels were measured as described previously (*Kawakami et al., 2011*) with minor modifications. After trypsin digestion of cells, the target peptide of each CYP isoform was absolutely quantified by LC-MS/MS. The expression levels of each CYP were quantified using previously described peptide standards (*Kawakami et al., 2011*).

## Measurement of cellular DNA

The cellular DNA content was measured to estimate the number of cells for CYP induction experiments. Following removal of Matrigel via aspiration, cells were washed once with phosphate-buffered saline (PBS) and any remaining Matrigel was removed by treating cells with Cell Recovery Solution (Corning) at 4°C for approximately 30 min. Thereafter, cells were washed once with PBS, and the cellular DNA content was determined using a DNA Quantity Kit (Cosmobio, Tokyo, Japan). To estimate the cell number from the DNA content, the correlation between these two parameters was determined using a dilution series of hCLiPs derived from each lot.

## qRT-PCR

Total RNA was isolated using an miRNeasy Mini Kit (QIAGEN, Venlo, The Netherlands). Reverse transcription was performed using a High-Capacity cDNA Reverse Transcription Kit (Life Technologies) according to the manufacturer's guidelines. cDNA was used for PCR with Platinum SYBR Green qPCR SuperMix UDG (Lifetechnologies). Expression levels of target genes were normalized against that of *ACTB* as an endogenous control. The primers used for qRT-PCR are listed in the following table.

## Primers for qRT-PCR

| Gene | Forward | Reverse |
|------|---------|---------|
| *ACTB* | ACTCTTCCAGCCTTCCTTCC | AGCACTGTGTTGGCGTACAG |
| *ALB* | GCAAGGCTGACGATAAGGAG | CCTAAGGCAGCTTGACTTGC |
| *TAT* | ATCTCTGTTATGGGGCGTTG | ACTAACCGCTCCGTGAACTC |
| *TTR* | ATCTCCCCATTCCATGAGC | CATTCCTTGGGATTGGTGAC |
| *TDO2* | GGTGGTTCCTCAGGCTATCA | TGTCGGGGAATCAGGTATGT |
| *G6PC* | CCTTGCTGCTCATTTTCCTC | TGTGGATGTGGCTGAAAGTT |
| *CYP1A2* | CCCCAAGAAATGCTGTGTCT | AGGGCTTGTTAATGGCAGTG |
| *CYP2B6* | GGGGCACTGAAAAAGACTGA | AGTTCTGGAGGATGGTGGTG |
| *CYP3A4* | ATTGGCATGAGGTTTGCTCT | CGGGTTTTTCTGGTTGAAGA |
| *EPCAM* | TGGACATAGCTGATGTGGCTTA | CCAGGATCCAGATCCAGTTG |
| *PROM1* | AGTCGGAAACTGGCAGATAGC | GGTAGTGTTGTACTGGGCCAAT |
| *CD24* | AGGCGCGGACTTTTCTTT | GATGCTGGGTGCTTGGAG |
| *GJB1* | CTGCTCTACCCTGGCTATGC | GTAGACGTCGCACTTGACCA |
| *THY1* | ACCTACACGTGTGCACTCCA | GCCCTCACACTTGACCAGTT |
| *ACTA2* | CTGTTCCAGCCATCCTTCAT | GGCAATGCCAGGGTACATAG |
| *VIM* | TCTGGATTCACTCCCTCTGG | GGTCATCGTGATGCTGAGAA |

## Immunocytochemistry (ICC)

The antibodies used for ICC are listed in the table below. Cells were fixed in chilled methanol (−30° C) on ice for 5 min. In some experiments, cells were fixed in 4% paraformaldehyde (PFA) (Wako, Osaka, Japan) at room temperature for 15 min and permeabilized by treatment with PBS containing 0.05% Triton X-100 for 15 min. Thereafter, cells were washed three times with PBS, incubated in Blocking One solution (Nacalai Tesque, Kyoto, Japan) at 4°C for 30 min, and labeled with primary antibodies at room temperature for 1 hr or at 4°C overnight. The primary antibodies were detected

using Alexa Fluor 488- or Alexa Fluor 594-conjugated secondary antibodies (Life Technologies). Nuclei were counterstained with Hoechst 33342 (Dojindo).

## Antibodies for ICC

| Antibody | Host animal | Catalog # | Dilution | Manufacturer | Fixation |
|---|---|---|---|---|---|
| CYP3A4 | Rabbit | Ab3572 | 1:200 | Abcam | Methanol |
| MRP2 | Mouse | Ab3373 | 1:200 | Abcam | Methanol |
| HNF4A | Rabbit | sc-8987 | 1:200 | Santa Cruz | 4% PFA |
| MDR1 | Rabbit | sc-53241 | 1:200 | Santa Cruz | Methanol |
| CYP2C | Mouse | sc-53245 | 1:200 | Santa Cruz | Methanol |
| CYP1A2 | Mouse | sc-53241 | 1:200 | Santa Cruz | Methanol |
| TTR | Rabbit | Ab75815 | 1:500 | Abcam | Methanol |

## IHC

The antibodies used for IHC are listed in the table below. Formalin-fixed paraffin-embedded (FFPE) tissue samples were prepared. Following dewaxing and rehydration, heat-induced epitope retrieval was performed by boiling specimens in ImmunoSaver (Nissin EM, Tokyo, Japan) diluted 1/200 at 98°C for 45 min. Endogenous peroxidase was inactivated by treating specimens with methanol containing 0.3% $H_2O_2$ at room temperature for 30 min. Thereafter, specimens were permeabilized with 0.1% Triton X-100, treated with Blocking One solution at 4°C for 30 min, and incubated with primary antibodies at room temperature for 1 hr or at 4°C overnight. Sections were stained using ImmPRESS IgG-peroxidase kits (Vector Labs, Burlingame, CA) and a metal-enhanced DAB substrate kit (Life Technologies), according to the manufacturers' instructions. Finally, specimens were counterstained with hematoxylin, dehydrated, and mounted.

FFPE tissue samples were used for fluorescence IHC unless otherwise stated. Following dewaxing and rehydration, heat-induced epitope retrieval was performed by boiling specimens in ImmunoSaver (Nissin EM) diluted 1/200 at 98°C for 45 min and then the following staining steps were performed. Fresh frozen tissue blocks prepared using Tissue-Tek O.C.T. Compound (Sakura Finetek, Tokyo, Japan) were used for CYP1A2 and CYP3A4 staining. Fresh frozen liver sections prepared using a cryostat (Leica) were fixed in chilled (−30°C) acetone (Wako) for 5 min, washed three times with PBS, permeabilized with 0.1% Triton X-100, and treated with Blocking One solution at 4°C for 30 min. Thereafter, specimens were incubated with primary antibodies at room temperature for 1 hr or at 4°C overnight and then stained with a mixture of an Alexa Fluor 488-conjugated antibody (Invitrogen) (1:500) and an Alexa Fluor 594-conjugated antibody (Invitrogen) (1:500) at room temperature for 1 hr. Stained sections were mounted using Vectashield mounting medium containing DAPI (Vector Laboratories).

## Antibodies for IHC

| Antibody | Host animal | Catalog # | Dilution | Manufacturer | Tissue type |
|---|---|---|---|---|---|
| CYP2C | Mouse | sc-53245 | 1:200 | Santa Cruz | FFPE/frozen |
| MDR1 | Rabbit | sc-53241 | 1:200 | Santa Cruz | FFPE |
| Human Mitochondria | Mouse | ab92824 | 1:1000 | Abcam | FFPE |
| Human TTR | Rabbit | ab75815 | 1:500 | Abcam | FFPE |
| GLUL | Rabbit | ab73593 | 1:1000 | Abcam | FFPE |
| Human CYP1A2 | Rabbit | BML-CR3130-0100 | 1:200 | Enzo | Frozen |
| Human CYP3A4 | Rabbit | BML-CR3340-0100 | 1:200 | Enzo | Frozen |

Liver repopulation assay using cDNA-uPA/SCID mice hCLiPs derived from three lots of cells were used. For lots FCL and JFC, primary cultured cells at D11–14 (P0-hCLiPs), cells passaged once (P1-hCLiPs), and cells passaged twice (P2-hCLiPs) were used. For lot FCL, P4-hCLiP transplantation was also performed. For lot DUX, P0-hCLiPs were used. After harvesting cells using TrypLE Express, 0.2–$1 \times 10^6$ cells/mouse were intrasplenically transplanted into 2–4 week-old cDNA-uPA/SCID mice (PhoenixBio Co., Ltd, Higashihiroshima, Japan) under isoflurane anesthesia. From 2 weeks after transplantation, 10 µl blood was retro-orbitally collected each week and the hALB concentration was measured using a Human Albumin ELISA Quantitation Kit (Bethyl, TX) or a Latex agglutination turbidimetric immunoassay with a BioMajesty analyzer (JCA-BM6050; JEOL, Tokyo, Japan). Livers were extracted at 8–11 weeks after transplantation and histologically analyzed. The transplantation experiments were approved by animal care committee.

## Liver repopulation assay using TK-NOG mice with P0-hCLiPs

FCL-P0-hCLiPs were used. Seven-week-old TK-NOG mice were obtained from the Central Institute of Experimental Animals (Kawasaki, Japan). One day after arrival at the National Cancer Center, mice were intraperitoneally injected with 10 mg/ml ganciclovir (Mitsubishi Tanabe Pharma Corporation, Osaka, Japan) at a dose of 10 µl/g body weight to induce thymidine kinase-mediated injury in host mouse hepatocytes. One week after injection, approximately 30 µl blood was obtained from the tail. Serum was separated and diluted 1/5 with PBS, and the serum ALT level was measured using a DRI-CHEM 3500 analyzer (Fujifilm, Tokyo, Japan). Mice with serum ALT levels of 500–1600 U/l were chosen as host animals for transplantation. At 1–3 days after ALT measurement, 0.4–$1 \times 10^6$ cells were intrasplenically transplanted into these mice under isoflurane anesthesia. From 2 weeks after transplantation, approximately 20 µl blood was collected each week from the tail and the hALB concentration was measured using a Human Albumin ELISA Quantitation Kit (Bethyl, Montgomery, TX). Livers were extracted at 8–10 weeks after transplantation and histologically analyzed. The transplantation experiments were approved by animal care committee.

## Liver repopulation assay using TK-NOG mice with P3-hCLiPs

For transplantation of FCL-P3-hCLiPs, EPCAM⁺ FCL-hCLiPs were magnetically sorted at the first passage (using P0 hCLiPs) by MACS cell sorting system using MidiMACS Separator (Miltenyi) using CD326 (EpCAM) MicroBeads (Miltenyi). These EPCAM+ cells were subjected to another two passages (P3 in total). During the culture at P3, we separated these cells to two groups, one with hepatic induction (P3 Hep-i(+)) and the other without hepatic induction (P3 Hep-i(-)). $1 \times 10^6$ cells/mouse were transplanted into GCV-treated TK-NOG mice as described before. Throughout this experiment, hCLiPs were cultured in a slightly different condition from other experiments (FBS concentration was 5% instead of 10%), but we do not think the obtained results were severely changed by this minor modification. Hepatic induction was conducted in SHM supplemented with 5%FBS and AC with 10 ng/ml hOSM (matrigel was not used). The transplantation experiments were approved by animal care committee.

## Estimation of RIs

Unless otherwise stated, RIs were estimated based on CYP2C positivity using image analysis software and a Keyence BZX-710 microscope. RIs in chimeric mice that were sacrificed to isolate primary hepatocytes were estimated based on magnetic bead separation, as described in the following section.

## Isolation of human hepatocytes from chimeric livers of cDNA-uPA/SCID mice

Hepatocytes were isolated from chimeric livers of cDNA-uPA/SCID mice at 10 weeks after transplantation of FCL-P1-hCLiPs, DUX-P0-hCLiPs, and JFC-P0-hCLiPs using a two-step collagenase perfusion method. To remove contaminating mouse hepatocytes, isolated cells were incubated with the 66Z antibody, which recognizes the surface of mouse hepatocytes, but not of human hepatocytes (*Yamasaki et al., 2010*). Cells were washed with DMEM containing 10% FBS and then incubated with Dynabeads M450-conjugated sheep anti-rat IgG (Dynal Biotech, Milwaukee, WI) for 30 min on ice. The tube was placed in a Dynal MPC-1 holder (Dynal Biotech) for 1–2 min to remove 66Z⁺

mouse hepatocytes. Human hepatocytes were collected as 66Z⁻ cells. 66Z⁺ and 66Z⁻ hepatocytes were counted using a hemocytometer before and after magnetic separation to estimate the repopulation efficiency and purity of human hepatocytes after separation, respectively.

## Culture of chimeric liver-derived human hepatocytes

Magnetically purified human hepatocytes were resuspended in SHM containing 2% FBS and seeded into a 24-well collagen I-coated plate. One day later, RNA was prepared from cells in some wells for microarray-based transcriptomic analysis. As a control, RNA was also prepared from hepatocytes isolated from the chimeric liver of a mouse transplanted with IPHHs (lot JFC) immediately after thawing the original cell suspension (kindly prepared by PhoenixBio Co., Ltd). Other hCLiP-derived hepatocytes were used for the CYP activity assay, as described above.

## Statistics

Data represent the mean ± SEM of independently repeated experiments or the mean ± SD of technical replicates in separate culture wells. Two groups were statistically compared using the Student's t-test, unless otherwise stated. Time-dependent alteration of gene expression was analyzed by the linear mixed models using IBM SPSS Statistics 23 (SPSS Inc, Chicago, IL). Group allocation (FBS or FAC), time (culture period [day]), and the interaction of group and time were included in the model as fixed effects. A p-value less than 0.05 was considered statistically significant.

## Accession numbers

Microarray transcriptome data are available with accession numbers GSE133776 (Reprogramming of primary human hepatocytes (PHHs) into hCLiPs); GSE133777 (Hepatic induction of hCLiPs); GSE133778 (Characterization of long term-cultured of hCLiPs); GSE133779 (Transcriptomic analysis of PHHs isolated from hCLiP-transplanted mouse chimeric liver). GSE133776-GSE133779 are included in Superseries GSE133797. Comparative analysis of IPHH and APHH transcriptome is available with an accession number GSE134672.

## Acknowledgements

We thank Ms. Ayako Inoue for technical help; Dr. Chise Tateno and her colleagues (PhoenixBio Co., Ltd) for assistance with the transplantation experiments, kindly providing chimeric liver samples repopulated with IPHHs (lot JFC), and valuable advice; Dr. Ben Z Stanger for generously permitting us to use the flow cytometry data of Rosa[YFP] mice; Drs. Taiji Yamazoe and Allyson J Merrell for critically reading the manuscript; and Drs. Luc Gailhouste and Yusuke Yamamoto for valuable advice. This research was supported in part by Grants-in-Aid from the Research Program on Hepatitis from Japan Agency for Medical Research and Development (AMED: 16fk0310512h0005 and 17fk0310101h0001, to TO), a grant from InterStem Co., Ltd (to TO), a Grant-in-Aid for Young Scientists B (16K16643, to TK.).

## Additional information

### Competing interests

Takahiro Ochiya: Received funding from Interstem Co. Ltd. The other authors declare that no competing interests exist.

### Funding

| Funder | Grant reference number | Author |
| --- | --- | --- |
| Japan Agency for Medical Research and Development | 16fk0310512h0005 | Takahiro Ochiya |
| Japan Agency for Medical Research and Development | 17fk0310101h0001 | Takahiro Ochiya |
| InterStem Co, Ltd | | Takahiro Ochiya |

| Japan Society for the Promotion of Science London | 16K16643 | Takeshi Katsuda |

The funders had no role in study design, data collection and interpretation, or the decision to submit the work for publication.

## Author contributions
Takeshi Katsuda, Conceptualization, Data curation, Formal analysis, Funding acquisition, Validation, Investigation, Visualization, Methodology, Writing—original draft, Writing—review and editing; Juntaro Matsuzaki, Yasuhiro Yamada, Data curation, Formal analysis, Investigation, Writing—review and editing; Tomoko Yamaguchi, Data curation, Validation, Investigation; Marta Prieto-Vila, Data curation; Kazunori Hosaka, Yoshimasa Saito, Investigation; Atsuko Takeuchi, Data curation, Investigation; Takahiro Ochiya, Conceptualization, Resources, Supervision, Funding acquisition

## Author ORCIDs
Takeshi Katsuda (ID) https://orcid.org/0000-0003-3960-033X
Juntaro Matsuzaki (ID) http://orcid.org/0000-0002-3204-5049
Takahiro Ochiya (ID) https://orcid.org/0000-0002-0776-9918

## Ethics
Animal experimentation: Animal experiments in this study were performed in compliance with the guidelines of the Institute for Laboratory Animal Research, National Cancer Center Research Institute. The protocol was approved by the Committee on the Ethics of Animal Experiments of National Cancer Center Research Institute (Permit Number: T14-015-E). All surgery was performed under isoflurane anesthesia, and every effort was made to minimize suffering.

## Decision letter and Author response
Decision letter https://doi.org/10.7554/eLife.47313.sa1
Author response https://doi.org/10.7554/eLife.47313.sa2

## Additional files

### Supplementary files
• Supplementary file 1. All the gene sets enriched in FAC cells compared with FBS cells at D14 of culture (assessed by GSEA).

• Supplementary file 2. All the gene sets enriched in FAC cells at D14 of culture compared with D1 hepatocytes (assessed by GSEA).

• Supplementary file 3. All the gene sets enriched in FBS cells at D14 of culture compared with D1 hepatocytes (assessed by GSEA).

• Supplementary file 4. Significantly enriched gene sets (Nom p<0.05) in Hep-i(+) cells compared with Hep-i(-) cells (assessed by GSEA).

• Supplementary file 5. Significantly enriched gene sets (Nom p<0.05) in Hep-i(-) cells compared with Hep-i(+) cells (assessed by GSEA).

• Supplementary file 6. Significantly enriched (NOM p<0.05) gene sets in hCLiP-chimera-derived hepatocytes in comparison with PHHs.

• Transparent reporting form

### Data availability
Microarray transcriptome data are available with accession numbers GSE133776 (Reprogramming of primary human hepatocytes (PHHs) into hCLiPs); GSE133777 (Hepatic induction of hCLiPs); GSE133778 (Characterization of long term-cultured of hCLiPs); GSE133779 (Transcriptomic analysis of PHHs isolated from hCLiP-transplanted mouse chimeric liver). GSE133776-GSE133779 are

included in Superseries GSE133797. Comparative analysis of IPHH and APHH transcriptome is available with an accession number GSE134672.

The following datasets were generated:

| Author(s) | Year | Dataset title | Dataset URL | Database and Identifier |
|---|---|---|---|---|
| Takeshi Katsuda, Takahiro Ochiya | 2019 | Reprogramming of primary human hepatocytes (PHHs) into hCLiPs | https://www.ncbi.nlm.nih.gov/geo/query/acc.cgi?acc=GSE133776 | NCBI Gene Expression Omnibus, GSE133776 |
| Takeshi Katsuda, Takahiro Ochiya | 2019 | Hepatic induction of hCLiPs | https://www.ncbi.nlm.nih.gov/geo/query/acc.cgi?acc=GSE133777 | NCBI Gene Expression Omnibus, GSE133777 |
| Takeshi Katsuda, Takahiro Ochiya | 2019 | Characterization of long term-cultured of hCLiPs | https://www.ncbi.nlm.nih.gov/geo/query/acc.cgi?acc=GSE133778 | NCBI Gene Expression Omnibus, GSE133778 |
| Takeshi Katsuda, Takahiro Ochiya | 2019 | Transcriptomic analysis of PHHs isolated from hCLiP-transplanted mouse chimeric liver | https://www.ncbi.nlm.nih.gov/geo/query/acc.cgi?acc=GSE133779 | NCBI Gene Expression Omnibus, GSE133779 |
| Takeshi Katsuda, Takahiro Ochiya | 2019 | Comparison between infant and adult primary human hepatocytes (PHHs) in terms of their responsiveness to FAC (FBS + A83-01 + CHIR99021) | https://www.ncbi.nlm.nih.gov/geo/query/acc.cgi?acc=GSE134672 | NCBI Gene Expression Omnibus, GSE134672 |

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
