## [Decision Letter]

Thank you for submitting your article "Generation of human hepatic progenitor cells with regenerative and metabolic capacities from primary hepatocytes" for consideration by *eLife*. Your article has been reviewed by three peer reviewers, one of whom is a member of our Board of Reviewing Editors, and the evaluation has been overseen by a Reviewing Editor and Anna Akhmanova as the Senior Editor. The following individual involved in review of your submission has agreed to reveal his identity: Bruce Wang (Reviewer #2).

The reviewers have discussed the reviews with one another and the Reviewing Editor has drafted this decision to help you prepare a revised submission.

Summary:

There is great interest in ex vivo sources of hepatocytes for transplantation or experimental studies. Derivation or expansion of these cells from ESCs or IPSCs or hepatocytes have not been satisfactory in terms of their ability to repopulate mice. Here, the authors explore a combination of factors plus FBS that causes the proliferation of infant but not adult hepatocytes and thus allows for the expansion of this population in vitro and in vivo. Upon addition of maturation media, there is also an increase in differentiation gene expression and a decrease of proliferation genes. This is important because it shows that the proliferating hepatocytes are indeed capable of returning to the well differentiated state. Moreover, there is good CYP expression, which allows for drug studies. Transplantation into a liver damage model shows that these cells can recover up to 90% of the recipient livers, although the effects are highly variable between animals. Overall, this is a compelling and balanced presentation of a new protocol for expansion and differentiation of hepatocytes from an infant hepatic source. This adds to the discussion about best ways to promote hepatocyte expansion.

The results are significant, and the work is done well, but there were concerns about cell purity, limits to passaging ability, and there needs to be more discussion about how the paper compares to other methods of expanding hepatocytes. Since their main reason to moving to young human cells is that they can be expanded more, we think it's crucial that they transplant the later passage cells.

Essential revisions:

The key requests that need to be addressed by experimental work are to transplant later passage cells and to do some purifications or elucidation of heterogeneity of cells in the culture.

1) There is no comment about the mechanisms or targets of the small molecules used and how this allows for proliferation ex vivo. At least a discussion about all of the relevant molecules is needed, as well as a comparison with other approaches, ie Hui et al.

2) One issue that consistently comes up in this paper is the heterogeneity in the cell culture samples. The authors should clarify whether this is due to contamination in their source hepatocytes with non-hepatocytes, or if the culture conditions generate heterogeneous cell fates. If due to contamination with non-hepatocytes, can the authors purify the cells before culturing? If they continue to find heterogeneity in their culture conditions despite starting with a pure hepatocyte population, it would be important to discuss why this might be occurring.

3) Related to the above point, the use of bulk gene expression profiling to make inferences about the cell types involved is not that definitive, and thus it would be better to minimize these inferences. For example, in subsection “Characterization of proliferating cells cultured in FAC” there is a lot of discussion about potential gene expression correlates of possible cell types, and it is not clear if this is true. It is not certain that TGFB signaling really says that there is a presence of fibroblastic LPCs in the culture. Single cell sequencing would be better for this. If single cell sequencing will not be done, then the discussion about how bulk sequencing can infer cell populations should be limited.

4) We believe that more transplantation experiments need to be performed in order to (A) understand why the RI is either very high or almost zero, and (B) understand if later passage cells will repopulate. This is the central issue with the first submission of the paper. For A), there needs to be an explanation for why there is a dichotomous result here. Otherwise it is hard to know what the actual RI is for these cells. Related to this, it is unclear how many passages can be performed while retaining hepatic differentiation potential. If the cells cannot be passaged many times, then it will be extremely difficult to generate enough cells for transplantation. Given that the in vitro expansion by passage 2 is limited, can the authors transplant cells after passage 4, when the cells are still relatively pure in vitro and have expanded more extensively. This is a significant issue with the paper, especially if there is going to be no mechanistic exploration of FAC.

5) The data are consistent with two recent publications showing that primary human hepatocytes can be maintained and expanded in vitro (Hu et al., 2018, and Zhang et al., 2018). In particular the paper from Zhang et al., is very comparable to the findings in this paper. Both papers use primary human hepatocytes cultured in 2D culture. The authors need to discuss their findings in the context of these recent publications. In particular, it would be very interesting to compare their culture conditions and results with that of Zhang et al., including the gene expression profiles and proliferative capacity.

We have also included the separate reviews for your consideration.

*Reviewer 1:*

There is great interest in ex vivo sources of hepatocytes for transplantation or experimental studies. Derivation or expansion of these cells from ESCs or IPSCs or hepatocytes have not been satisfactory in terms of their ability to repopulate mice. Here, the authors explore a combination of factors plus FBS that causes the proliferation of infant but not adult hepatocytes and thus allows for the expansion of this population in vitro and in vivo. Upon addition of maturation media, there is also an increase in differentiation gene expression and a decrease of proliferation genes. This is important because it shows that the proliferating hepatocytes are indeed capable of returning to the well differentiated state. Moreover, there is good CYP expression, which allows for drug studies. Transplantation into a liver damage model shows that these cells can recover up to 90% of the recipient livers, although the effects are highly variable between animals. Overall this is a compelling and balanced presentation of a new protocol for expansion and differentiation of hepatocytes from an infant hepatic source. This adds to the conversation about best ways to promote hepatocyte expansion.

Essential revisions:

There is no comment about the mechanisms or targets of the small molecules used and how this allows for proliferation ex vivo. At least a discussion about this is needed, and a comparison with other approaches, ie Hui et al.

The use of bulk gene expression profiling to make inferences about the cell types involved is not that definitive, and thus i would minimize this. For example, subsection “Characterization of proliferating cells cultured in FAC” there is a lot of discussion about potential gene expression correlates of possible cell types, and I think it is not clear if this is true. I am not sure that TGFB signaling in hips really says that there is a presence of fibroblastic LPCs in the culture. I think single cell sequencing would be better for this. If single cell seq will not be done, then I think the discussion about how bulk seq can infer cell populations should be limited.

It is unclear how many passages can be performed while retaining hepatic differentiation potential. If the cells cannot be passage many times, then it will be extremely difficult to generate enough cells for transplantation. I think 5 passages is not that significant. Please comment on this.

I believe more transplantation experiments need to be performed in order to understand why the RI is either very high or almost zero. There needs to be an explanation for why there is a dichotomous result here. Otherwise it is hard to know what the actual RI is for these cells, especially those that are in higher passages beyond P1 or P2. This is a significant issue with the paper, especially if there is going to be no mechanistic exploration of FAC.

*Reviewer 2:*

In this manuscript, Katsuda et al., follow up their previous rodent study by describing culture conditions that allow for the expansion of primary human hepatocytes. They showed that while adult primary human hepatocytes have limited proliferative capacity in their culture conditions, hepatocytes from young human livers can expand efficiently, and over multiple passages. They show extensive characterization of these in vitro expanded cells, including transplantation into mouse models of liver failure to demonstrate the capacity of these cells to function as hepatocytes in vivo.

Overall, this is a well written manuscript, with carefully conducted studies and detailed characterization. It addresses an important bottleneck in the field of liver regenerative medicine, the ability to maintain and expand hepatocytes in vitro for either in vitro drug testing or in vivo transplantation. However, I have a few concerns about the manuscript that need to be addressed prior to publication. My specific comments are below.

1) Their data is consistent with two recent publications showing that primary human hepatocytes can be maintained and expanded in vitro (Hu et al., 2018, and Zhang et al., 2018). In particular the paper from Zhang et al. is very comparable to the findings in this paper. Both papers use primary human hepatocytes cultured in 2D culture. The authors need to discuss their findings in the context of these recent publications. In particular, it would be very interesting to compare their culture conditions and results with that of Zhang et al., including the gene expression profiles of and proliferative capacity.

2) One issue that consistently comes up in this paper is the heterogeneity in their cell culture samples. The authors should clarify whether this is due to contamination in their source hepatocytes with non-hepatocytes, or if the culture condition generates heterogeneous cell fates. If due to contamination with non-hepatocytes, can they purify the cells before culturing? If they continue to find heterogeneity in their culture conditions despite starting with a pure hepatocyte population, it would be interesting to separate the different resulting cell populations in vitro and analyze them separately.

3) In transplantation studies, the authors state in the text that repopulation capacity declines as culture period increased, but this is not clearly seen in figure 5C. Since the authors think the uPA/SCID mice have lower engraftment efficiency, it would be nice to see transplantation of P1 and P2 cells in the TK-NOG mice.

4) The in vitro expansion by passage 2 is limited. Can they transplant cells after passage 4, when the cells are still relatively pure in vitro and have expanded more extensively?

5) Characterization of the gene expression and enzymatic function in vitro is well done. Goes farther than similar articles in this respect and provides good rational for these cells to be used in in vitro drug metabolism studies.

*Reviewer 3:*

Katsuda et al., showed the efficacy of their generated hCLiPs in vitro and in vivo as well as previous rat hepatocyte-derived CLiPs. This progress shown in this paper is important. I have a few concerns and requests about this paper.

1) The authors also expected that hCLiPs would be used in the liver cell transplantation. On the other hand, it was difficult to generate hCLiPs from adult human hepatocytes. As you know, infant liver donor is shortage actually. The authors need to assess the mechanism of difficulties in the generation of CLiPs from adult hepatocytes.

2) The authors showed that the combination of defined chemicals (A-83-01 and CHIR99021) and FBS is essential for hCLiPs-generation. The authors are requested to explain the functions of Y(Y-27632) which is not included.

---

## [Author Response]

[…] The results are significant, and the work is done well, but there were concerns about cell purity, limits to passaging ability, and there needs to be more discussion about how the paper compares to other methods of expanding hepatocytes. Since their main reason to moving to young human cells is that they can be expanded more, we think it's crucial that they transplant the later passage cells. […]Reviewer 1:There is great interest in ex vivo sources of hepatocytes for transplantation or experimental studies. Derivation or expansion of these cells from ESCs or IPSCs or hepatocytes have not been satisfactory in terms of their ability to repopulate mice. Here, the authors explore a combination of factors plus FBS that causes the proliferation of infant but not adult hepatocytes and thus allows for the expansion of this population in vitro and in vivo. Upon addition of maturation media, there is also an increase in differentiation gene expression and a decrease of proliferation genes. This is important because it shows that the proliferating hepatocytes are indeed capable of returning to the well differentiated state. Moreover, there is good CYP expression, which allows for drug studies. Transplantation into a liver damage model shows that these cells can recover up to 90% of the recipient livers, although the effects are highly variable between animals. Overall this is a compelling and balanced presentation of a new protocol for expansion and differentiation of hepatocytes from an infant hepatic source. This adds to the conversation about best ways to promote hepatocyte expansion.

First of all, we sincerely appreciate the reviewer for carefully reading our manuscript and providing us insightful comments. As indicated in the point-by-point responses below, we have taken all of your comments and suggestions into account in the revision of this manuscript. We hope that you will consider our responses as satisfactory.

Essential revisions:There is no comment about the mechanisms or targets of the small molecules used and how this allows for proliferation ex vivo. At least a discussion about this is needed, and a comparison with other approaches, ie Hui et al.

To address the request by the reviewer, we added two paragraphs to the Discussion section. In the third paragraph, we discussed the possible mechanism underlying small molecule-mediated induction of proliferation of PHHs, and in the fourth paragraph we compared our study and the recently published similar works (Fu et al., 2018; Hu et al., 2018; Kim et al., 2018; Zhang et al., 2018). In addition to these discussions, we also performed microarray analysis to compare the responsiveness to FAC of IPHHs (infant PHHs) and APHHs (adult PHHs) (Figure 3, and its supplementary figures). The results suggested that IPHHs are more sensitive to CHIR99021 than adult PHHs, since Wnt signaling was more efficiently upregulated in IPHHs.

The use of bulk gene expression profiling to make inferences about the cell types involved is not that definitive, and thus i would minimize this. For example, in subsection “Characterization of proliferating cells cultured in FAC” there is a lot of discussion about potential gene expression correlates of possible cell types, and I think it is not clear if this is true. I am not sure that TGFB signaling in hips really says that there is a presence of fibroblastic LPCs in the culture. I think single cell sequencing would be better for this. If single cell seq will not be done, then I think the discussion about how bulk seq can infer cell populations should be limited.

As per the comment, we deleted the part which this reviewer pointed out. Accordingly, we also deleted the corresponding GSEA data.

It is unclear how many passages can be performed while retaining hepatic differentiation potential. If the cells cannot be passaged many times, then it will be extremely difficult to generate enough cells for transplantation. I think 5 passages is not that significant. Please comment on this.

As shown in Figure 6, we described that hCLiPs can be differentiated into functional hepatocytes until passage 5. At the end of the 5th passage, we can obtain 1.2 ± 0.35 x 10^8^, 3.3 ± 0.44 x 10^7^ and 2.0 x 10^6^ hCLiPs from one starting PHH for lots FCL, DUX and JFC, respectively. We believe these data suggest substantial benefit of using hCLiPs, given the limited availability of PHHs.

Also please note that the in vitro proliferative capacity of hCLiPs is much higher than that reported in the recently reported similar papers: FCL and DUX hCLIPs underwent 2.9 x 10^13^- and 2.2 x 10^12^-fold expansion at the end of P10 culture, and JFC hCLiPs underwent 2.0 x 10^6^-fold expansion at the end of P5. In contrast, Zhang et al. reported that ProliHH achieved 1,000- to 10,000-fold expansion at P8; this fold-expansion is nearly equivalent to that of hCLiPs at P3 (Zhang et al., 2018). Kim et al. reported that hCdHs were passaged at 1:3-1:5 in each passage, which corresponds to 5.9 x 10^4^ – 9.7 x 10^6^-fold expansion at P10; this fold-expansion is nearly equivalent to that of hCLiPs at P4-5 (Kim et al., 2018).

I believe more transplantation experiments need to be performed in order to understand why the RI is either very high or almost zero. There needs to be an explanation for why there is a dichotomous result here. Otherwise it is hard to know what the actual RI is for these cells, especially those that are in higher passages beyond P1 or P2. This is a significant issue with the paper, especially if there is going to be no mechanistic exploration of FAC.

Given that three lots of hCLiPs consistently exhibited the dichotomous results in transplantation experiment, we think this is the nature of hCLiPs. Although we do not have any evidence, it seems to us that the high/low RI is determined by the initial engraftment efficiency. Whether this idea is correct or not, and if it is correct, what molecular mechanism specifically determines the successful engraftment must be addressed in future studies.

This reviewer requested to increase the “n” by repeating the same transplantation experiment. We acknowledge that this is an important comment. However, the editor suggested us to rather perform repopulation experiments using hCLiPs at later passages. Following the editor’s suggestion, we performed repopulation study using FCL^-^P3-hCLiPs and FCL^-^P4-hCLiPs, which were transplanted to TK-NOG and cDNA-uPA/SCID mice, respectively. As you can see in Figure 8—figure supplement 1, repopulation efficiency of hCLiPs at late passages is overall much lower than that at early passage (P0-P1). We sincerely accept that the low repopulation efficiency of late passaged hCLiPs is one of the critical limitations in our study.

Reviewer 2:[…] Overall, this is a well written manuscript, with carefully conducted studies and detailed characterization. It addresses an important bottleneck in the field of liver regenerative medicine, the ability to maintain and expand hepatocytes in vitro for either in vitro drug testing or in vivo transplantation. However, I have a few concerns about the manuscript that need to be addressed prior to publication. My specific comments are below.

First of all, we appreciate your time and effort in reviewing our paper. Your valuable comments provided significant help in improving our manuscript. We have modified our former version to address your requests. As indicated in the point-by-point responses below, we have taken all of your comments and suggestions into account in the revision. We hope that you will consider our responses as satisfactory.

1) Their data is consistent with two recent publications showing that primary human hepatocytes can be maintained and expanded in vitro (Hu et al., 2018, and Zhang et al., 2018). In particular the paper from Zhang et al., is very comparable to the findings in this paper. Both papers use primary human hepatocytes cultured in 2D culture. The authors need to discuss their findings in the context of these recent publications. In particular, it would be very interesting to compare their culture conditions and results with that of Zhang et al., including the gene expression profiles of and proliferative capacity.

We added two paragraphs to Discussion section. In the third paragraph, we discussed the possible mechanism underlying small molecule-mediated induction of proliferation of PHHs, and in the fourth paragraph we compared our study with the recently published similar studies (Fu et al., 2018; Hu et al., 2018; Kim et al., 2018; Zhang et al., 2018).

2) One issue that consistently comes up in this paper is the heterogeneity in their cell culture samples. The authors should clarify whether this is due to contamination in their source hepatocytes with non-hepatocytes, or if the culture condition generates heterogeneous cell fates. If due to contamination with non-hepatocytes, can they purify the cells before culturing? If they continue to find heterogeneity in their culture conditions despite starting with a pure hepatocyte population, it would be interesting to separate the different resulting cell populations in vitro and analyze them separately.

We do not think it is feasible to purify cells before culturing, because almost none of LPC markers or an NPC marker are expressed in the initial PHH source (Figure 2—figure supplement 2B, Author response image 1). To investigate the population heterogeneity of the starting PHHs, we performed flow cytometry using PHHs immediately after thawing the cryopreserved cell stocks. A hepatic surface marker DPP4/CD26 expression was detected in each of the tested cells (except JFC PHHs, in which this antigen was accidentally not tested), thereby confirming that surface antigens of these cells were at least partly preserved even after a freeze/thaw cycle. We tested the surface expression of LPC markers EPCAM, PROM1, CD24, CD44 and ITGA6. Except for ITGA6, none of these surface proteins were detected in these cells. ITGA6 expression in PHHs, including APHHs (HC5-25, derived from a 56 year-old donor), was surprising to us. Thus, we interrogated whether this slight ITGA6 expression in PHHs would be caused by LPC contamination or a nature of primary hepatocytes, using a mouse lineage tracing model. We genetically labeled hepatocytes with YFP by injecting AAV-TBG-Cre to a Rosa^YFP^ adult mouse, harvested its hepatocytes, and performed flow cytometry. While YFP^+^ hepatocytes were negative for any of Epcam, Prom1 and Cd24, we observed a slight signal of Itga6 (Figure 2—figure supplement 2C), just like in the case of human hepatocytes. Thus, we conclude that the slight expression of ITGA6 in PHHs does not reflect LPC contamination. As pointed out by the reviewer, we also investigated the contamination of fibroblastic cells with the use of general fibroblastic marker, THY1/CD90. THY1 expression in FCL hCLiPs was negligible (Author response image 1), indicating that our starting cells are mostly composed of PHHs.

Nonetheless, we know that morphologically fibroblastic cells emerge during culture, and thus we carefully followed up the cultured cells using THY1 antibody in combination with LPC markers. After primary expansion of hCLiPs (P0), we confirmed that a certain population of cells were THY1^+^. Without genetic lineage tracing, which is practically unfeasible with PHHs, we cannot determine whether these cells were derived from minimally-contaminated fibroblasts or cells de-differentiated from PHHs per se. Thus, we refrained from discussing much about the cellular origin of these cells in the manuscript.

Then, we sorted LPC marker^+^THY1^-^ putative LPC fractions and LPC marker^-^THY^+^ fractions, and compared their gene expression profiles using a panel of genes (LPC markers, hepatic markers and fibroblastic markers). As expected, LPC marker^+^THY1^-^ populations were rich in LPC/hepatic cells, while LPC marker^-^THY1^+^ populations were rich in fibroblastic cells (Figure 7B). We then continued culture of these cells for another two weeks and followed up their gene expression profiles. The sorted populations still maintained their characteristics partly, but LPC marker^-^THY1^+^ cells again emerged in LPC marker^+^THY1^-^ fractions, suggesting that LPC marker^-^THY1^+^ cells which were not fully excluded by sorting grew in these fractions in the following culture. Moreover, we noted the substantial reduction in the expression of hepatic genes in LPC marker^+^THY1^-^ fractions when compared with P0 cells (Figure 7—figure supplement 2). Thus, we conclude that surface marker-based selection can temporarily enrich LPC/hepatic cells, but the decline of hepatic/LPC phenotype in hCLiPs is unavoidable at least in the present sorting strategy and culture condition.

3) In transplantation studies, the authors state in the text that repopulation capacity declines as culture period increased, but this is not clearly seen in figure 5C. Since the authors think the uPA/SCID mice have lower engraftment efficiency, it would be nice to see transplantation of P1 and P2 cells in the TK-NOG mice.

In our pilot study, we transplanted FCL^-^hCLiPs which were enriched for EPCAM expression by MACS at the first passage (namely, using P0 cells) and subsequently passaged twice (thus, P3 in total). Since we supposed that the reduction of repopulative capacity of hCLiPs in the later passages was largely due to the loss of hepatic phenotype, we also prepared hCLiPs with hepatic induction (*Hep*-i(+) cells). The data indicated that *Hep*-i(+) hCLiPs had higher repopulative capacity than *Hep*-i(-) cells. However, the repopulative capacity of *Hep*-i(+)-P3-hCLiPs was still much lower than P0-hCLiPs, highlighting the difficulty for hCLiPs to retain their repopulative capacity after extended culture periods.

By the way, in this experiment (which was a part of our pilot study), hCLiPs were cultured in a slightly different condition (FBS concentration was 5% instead of 10%. But we do not think that the obtained results would be severely changed, so we provided the data in Figure 8—figure supplement 1C.

4) The in vitro expansion by passage 2 is limited. Can they transplant cells after passage 4, when the cells are still relatively pure in vitro and have expanded more extensively?

We transplanted FCL^-^P4-hCLiPs to cDNA-uPA/SCID mice (n = 3). Although each mouse exhibited increase in blood hALB levels, its increasing rate was much slower than that of highly repopulated mice which were transplanted with P0- or P1-FCL^-^hCLiPs (Figure 8—figure supplement 1A). CYP2C IHC indicated that RIs were < 1% for each of the P4-hCLiP-transplanted animals 8 weeks after transplantation (Figure 8—figure supplement 1B). Because we observed very few foci formed in these mice, it seems that engraftment of hCLiPs at later passages were much less efficient than that at earlier passages.

On the other hand, we caution reviewers about the interpretation of the results regarding in vitro cellular expansion and repopulation. First, we noted that fold expansion of hCLiPs at P0-P2 are comparable to that of ProliHH at P4-P6 (Zhang et al., 2018). When comparing our study with Zhang et al’s one, it seems more appropriate to use the term “fold expansion” (from the initial cell number) rather than “passage number” to indicate the age of cultured cells. According to Figure S5A in Zhang et al.’s study, “Early ProliHH (P4)” and “Late ProliHH (P6)” underwent approximately 400- and 1000- fold expansion from the initial cell number (also note that their passage counting is different from ours: P4 and P6 in their counting system correspond to P3 and P5 based on our passage counting) (Author response image 2). FCL hCLiPs underwent > 40- fold expansion during primary culture (P0), approximately 1000-fold expansion at the endpoint of P1 (FCL^-^hCLiPs expand approximately 25-fold during P1 culture), and 6.4 x 10^6^-fold expansion at the end of P4 (Author response image 1). Thus, to our surprise, the fold expansion of our FCL P1 hCLiPs is comparable to that of Zhang et al.’s “Late ProliHH”. In our study, JFC-hCLiPs exhibited obviously slower growth rate compared to FCL^-^ and DUX-hCLiPs (Figure 6A). Nonetheless, we confirmed that JFC-hCLiPs at P2, which exhibited high RIs in 2 of 3 transplanted mice, underwent 456 ± 156-fold expansion (n = 2) at the end of P2 (see Author response image 2), which is comparable to that achieved by ProliHH at P3-P5. We mentioned this issue concisely in the Results section.

**Author response image 2. respfig2:** Comparison between ProliHHs and hCLiPs in terms of in vitro growth rates. (**A**) According to Figure S5A in Zhang et al., 2018, Early ProliHH and Late ProliHH underwent approximately 200 and 1000 fold expansion from the starting cell number. Note that our passage counting manner is different from theirs. (**B**) FCL hCLiPs underwent > 40-fold expansion during primary culture (P0), approximately 1000-fold expansion at P1, and 6.4 x 106-fold expansion at P4. It should be noted that our P1 cells correspond to Zhang et al.’s P6 cells in terms of fold expansion form the initial cell number.

Second, we noted that the repopulation rate (speed) is very different between hCLiPs and ProliHH. When hCLiPs achieved high RI, blood hALB levels reached 1 mg/ml as early as at 30 days post transplantation (Author response image 2), whereas ProliHH even at early passage (P4) reached this level approximately at 120 days post transplantation; they reached approximately 30-40 μg/ml (estimation from the graph) at 30 days post transplantation (see Author response image 3). This finding implies differences in repopulation modes between hCLiPs at early passage and ProliHH at both early and late passages. Another finding was that when we adjusted the range of x-axis to the 0-180 days post transplantation (same as ProliHH data), repopulation rate of FCL^-^P4-hCLiPs was comparable with that of Late ProliHH. Since FCL^-^P4-hCLiP’s hALB increasing rate was much slower than we expected based on our previous observation with P0- or P1-hCLiPs, we quitted this experiment at the standard endpoint of our experiment (8 weeks post transplantation). But, if we had extended this transplantation study until 150 days, it could be possible for P4 hCLiPs to achieve similar RI levels as ProliHH. However, this comparative investigation includes uncertain assumption/imagination which we are afraid to be “over-speculation”, and thus we did not include this discussion in our manuscript.

**Author response image 3. respfig3:** Comparison between ProliHHs and hCLiPs in terms of their repopulation efficiency. (A)Repopulation experiments in FRG mice using ProliHHs at early (P4) and late (P6) passages. Data are adapted from Supplementary Figure S6G (Zhang et al., 2018).(B)hCLiP (lot FCL) repopulation experiment data (cDNA-uPA/SCID) at early (P0 and P1) and at late (P4) passages are adapted from Figure 6—figure supplement 1C of the present study. JFC (P0, P1, P2) blood hALB data are adapted from Figure 6A in the present study. We omitted data for individuals without substantial repopulation from hCLiPs at early passages and added extended the range of x-axis, to make the comparison with our data and Zhang et al’s data easier.

5) Characterization of the gene expression and enzymatic function in vitro is well done. Goes farther than similar articles in this respect and provides good rational for these cells to be used in in vitro drug metabolism studies.

We are pleased to have this comment.

Reviewer 3:Katsuda et al., showed the efficacy of their generated hCLiPs in vitro and in vivo as well as previous rat hepatocyte-derived CLiPs. This progress shown in this paper is important. I have a few concerns and requests about this paper.

First of all, we wish to express our sincere thanks for your critical reading of our manuscript and your thoughtful comments and suggestions. As indicated in the point-by-point responses below, we have taken all of the comments and suggestions into account in our revision of this manuscript. We have provided additional data to support our conclusions, and thus, we hope that you will consider our responses to be satisfactory.

1) The authors also expected that hCLiPs would be used in the liver cell transplantation. On the other hand, it was difficult to generate hCLiPs from adult human hepatocytes. As you know, infant liver donor is shortage actually. The authors need to assess the mechanism of difficulties in the generation of CLiPs from adult hepatocytes.

To investigate the difference in the responsiveness to FAC of IPHHs and APHHs, we compared their transpcriptome by microarray analysis. As described in the new subsection “Comparison of IPHHs and APHHs in terms of their responsiveness to FAC” (data are provided in Figure 3), it was suggested that the lower proliferative capacity of APHHs might be partly explained by their lower responsiveness to Wnt signaling.

2) The authors showed that the combination of defined chemicals (A-83-01 and CHIR99021) and FBS is essential for hCLiPs-generation. The authors are requested to explain the functions of Y(Y-27632) which is not included.

We performed a mini-screen using all possible combinations of Y, A and C in 10% FBS-supplemented SHM, and found that the combination of AC rather than YAC allows PHHs to proliferate most efficiently. We do not have explanation for this observation, and thus we did not provide much discussion about this finding in the manuscript.